# Toll-like receptor 2 expression on c-kit$^+$ cells tracks the emergence of embryonic definitive hematopoietic progenitors

Jana Balounová[1,2,11], Iva Šplíchalová[1,3,11], Martina Dobešová[1], Michal Kolář[4], Karel Fišer [5], Jan Procházka[2], Radislav Sedlacek[2], Andrea Jurisicova[6], Hoon-ki Sung[7], Vladimír Kořínek[8], Meritxell Alberich-Jorda[9], Isabelle Godin [10] & Dominik Filipp [1*]

Hematopoiesis in mammalian embryos proceeds through three successive waves of hematopoietic progenitors. Since their emergence spatially and temporally overlap and phenotypic markers are often shared, the specifics regarding their origin, development, lineage restriction and mutual relationships have not been fully determined. The identification of wave-specific markers would aid to resolve these uncertainties. Here, we show that toll-like receptors (TLRs) are expressed during early mouse embryogenesis. We provide phenotypic and functional evidence that the expression of TLR2 on E7.5 c-kit$^+$ cells marks the emergence of precursors of erythro-myeloid progenitors (EMPs) and provides resolution for separate tracking of EMPs from primitive progenitors. Using in vivo fate mapping, we show that at E8.5 the *Tlr2* locus is already active in emerging EMPs and in progenitors of adult hematopoietic stem cells (HSC). Together, this data demonstrates that the activation of the *Tlr2* locus tracks the earliest events in the process of EMP and HSC specification.

[1] Laboratory of Immunobiology, Institute of Molecular Genetics of the Czech Academy of Sciences, Prague, Czech Republic. [2] Czech Centre for Phenogenomics & Laboratory of Transgenic Models of Diseases, Institute of Molecular Genetics of the Czech Academy of Sciences, Prague, Czech Republic. [3] Department of Cell Biology, Faculty of Science, Charles University, Prague, Czech Republic. [4] Laboratory of Genomics and Bioinformatics, Institute of Molecular Genetics of the Czech Academy of Sciences, Prague, Czech Republic. [5] Childhood Leukaemia Investigation Prague, 2nd Faculty of Medicine, Charles University, Prague, Czech Republic. [6] Samuel Lunenfeld Research Institute, Mount Sinai Hospital, Toronto, ON, Canada. [7] The Hospital for Sick Children Research Institute, Toronto, ON, Canada. [8] Laboratory of Cell and Developmental Biology, Institute of Molecular Genetics of the Czech Academy of Sciences, Prague, Czech Republic. [9] Laboratory of Hematooncology, Institute of Molecular Genetics of the Czech Academy of Sciences, Prague, Czech Republic. [10] Institute Gustave Roussy PR1, INSERM U1170, Villejuif, France. [11] These authors contributed equally: Jana Balounová, Iva Šplíchalová. *email: dominik.filipp@img.cas.cz

Mammalian hematopoiesis proceeds through three successive waves of progenitors. The first, referred to as the primitive wave, arises in the yolk sac (YS) at E7.25 and consists of progenitors of primitive nucleated erythrocytes (EryP) and megakaryocytes (Mk)[1–3]. Monopotent progenitors of macrophages (MFs) are also considered to be part of this wave[4,5]. At E8.5, a second, transient definitive wave of erythro-myeloid progenitors (EMPs) and lympho-myeloid progenitors (LMPs) emerges from the YS hemogenic endothelium (HE)[6–8]. Ultimately, around E10.5, hematopoietic stem cells (HSCs) emerge from the HE of the ventral floor of the dorsal aorta in the aorta–gonad–mesonephros region (AGM)[9,10]. Commencing from E9.5, EMPs from YS and other hemogenic tissues[11] and at E10.5 HSCs from AGM, start to seed and expand in the forming fetal liver (FL). Subsequently, these progenitors migrate and differentiate to tissue-resident MFs, with the latter populating the bone marrow (BM), spleen, or thymus[4,6,12–14].

Hematopoietic commitment in YS and AGM occurs independently of each other via the endothelial to hematopoietic transition (EHT) of *Runx1* expressing cells, and is traceable by the appearance of CD41 on the surface of c-kit[+] cells[8,15–17]. Due to differences in the timing of their appearance, lineage potential and combinatorial dependency on developmental factors, such as *Runx1*, *c-myb*, and *Notch*, the progenitors of the primitive erythropoiesis are considered to be distinct from EMPs and HSCs which exhibit definitive hematopoietic potential[18]. This is in agreement with the observation that primitive erythropoiesis occurs in *Runx1* deficient embryos, although EMPs, HSCs, and also MFs are absent[8,19,20]. However, other studies have suggested that these hematopoietic waves not only share their progenitors but also phenotypic markers, such as c-kit and CD41[3,8,21]. Due to low temporal resolution, lineage-tracing experiments that employ *Tie2*, *Runx1*, and *c-Kit* reporters have failed to track the separate emergence of primitive versus EMP-derived MFs[22–24]. Thus, identification of additional surface markers would be vital in revealing the developmental and functional relationship between hematopoietic waves.

Toll-like receptors (TLRs) recognize various structures of microbes and are crucial for triggering immune responses to infections[25,26]. TLR stimulation of adult BM HSCs during infection redirects BM hematopoiesis toward the increased production of myeloid cells, demonstrating their role in hematopoietic homeostasis under inflammatory conditions[27–29]. So far, only a few studies have analyzed the expression of TLRs in embryonic development[30–32], leaving the ontogeny of TLR expression in pre-circulation embryos unknown.

We show here that TLR2 is expressed on E7.5 c-kit[+] YS cells, which co-express the hematopoietic emergence markers *Runx1* and CD41 and exhibit the functional attributes of EMPs. In addition, E8.5 TLR2[+] c-kit[+] EMPs respond to TLR2 stimulation in a *MyD88*-dependent manner. By genetic labeling of progenitors with active *Tlr2*-regulatory elements, we have determined their contribution to definitive hematopoietic lineages. Altogether, co-expression of TLR2 and c-kit serves as a reliable marker to track the emergence of EMPs and HSCs as well as the tracing of their origin, emergence, fate, lineage potential, and function.

## Results

### E7.5 YS-derived TLR2[+]c-kit[+] cells are emerging precursors of EMPs.
We have previously shown that TLRs are expressed in E10.5 YS-derived MFs[33]. After deeper analysis, we detected the transcripts of *Tlrs* and their adaptors already at E7.5 (Supplementary Fig. 1a). At this time point, TLR protein expression, exemplified by TLR2, showed a scattered pattern of distribution across the YS. Anatomically, TLR2[LOW] cells were most abundant in the YS and posterior primitive streak (PPS), where cells undergo epithelial to mesenchymal transition (Fig. 1a; Supplementary Movie 1).

The localization of TLR2[+] cells within the PPS and YS suggests an association with early hematopoiesis. To focus exclusively on embryonically derived cells, we used crosses between wild-type (*Actb*[wt/wt]) and actin-β[EGFP] (*Actb*[EGFP/wt]) transgenic mice to distinguish cells of maternal and embryonic origin based on their EGFP expression (Supplementary Fig. 1b). This data confirmed that maternal MFs at ~E8.5–9.5 are replaced by embryo-derived ones[5] and that TLR2[+]CD11b[-]CD45[-] cells, already present at E7.5, are indeed of embryonic origin.

At E7.5, a large proportion of embryonic *Actb*[EGFP]TLR2[+]CD45[-] cells also co-expressed c-kit, a hallmark of hematopoietic progenitors. TLR2[+]c-kit[+] cells were largely restricted to YS, with significantly lower numbers in the embryo proper (EP) (Fig. 1b). Importantly, only TLR2[+]c-kit[+] cells from YS, but not those from EP, were positive for the early hematopoietic marker, CD41[3] (Fig. 1c). Similarly, TLR2[+]c-kit[+] progenitors isolated from YS, but not EP, expressed mRNA encoding *Runx1*, an obligatory transcription factor (TF) required for the emergence of definitive hematopoiesis (Fig. 1d). *Runx1* was also expressed by the YS-derived TLR2[-]c-kit[+] population, which was also positive for CD41 (Fig. 1c). This is consistent with the emergence of hematopoietic progenitors exclusively among c-kit[+] cells in the YS[8].

Next, we tested whether the expression of TLR2 on E7.5 c-kit[+] cells marks progenitors with an early commitment to a hematopoietic fate. Using *Actb*[EGFP] embryos, we gated on YS-derived embryonic TLR2[+] or TLR2[-] cells and analyzed the kinetics of their c-kit and CD45 expression. At E7.5, TLR2[+] cells separated into two subsets: one expressing high levels of c-kit while the other was c-kit[-] (Fig. 1e, top panels). During the specification of the TLR2[+] subset toward the CD45[+] cells, we observed three stages: (i) at E8.5, the c-kit[-] subset increased its c-kit expression with virtually all TLR2[+] cells becoming c-kit[+]; (ii) at E9.75, TLR2[+]c-kit[+]CD45[-] cells acquired CD45 surface expression; and (iii) concomitantly downregulated c-kit. In contrast, the TLR2[-] phenotype was associated with low/negative c-kit expression. However, despite the upregulation of c-kit expression in some cells at E8.5–9.5, the entire TLR2[-] subset remained CD45 negative (Fig. 1e, bottom panels).

Consistent with the observed transition to hematopoietic fate, TLR2[+]c-kit[+] cells expressed higher levels of endothelial, hematopoietic, and myeloid markers as well as TFs typical for EHT in comparison with the TLR2[-]c-kit[+] subset (Supplementary Fig. 1c, d). During their maturation toward the CD45[+] stage, the expression of myeloid markers increased while endothelial markers such as *Tie2* and *Cd31* were downregulated, a distinctive feature of EHT. It is of note, that *Gata1*, a TF required for the generation of primitive erythroblasts[34], was the only gene which was preferentially expressed in TLR2[-]c-kit[+] cells (Supplementary Fig. 1d). These data imply that in E7.5–10.5 YS, the process of hematopoietic maturation toward CD45[+] stage is restricted to TLR2[+]c-kit[+] progenitors.

To establish if the developmental progression of E7.5 TLR2[+]c-kit[+] progenitors is cell autonomous, we assessed their hematopoietic potential in a clonogenic assay at days 5 and 12 (Fig. 1f; Supplementary Fig. 2a, b). After 5 days of culture, only c-kit[+] cells gave rise to visible colonies with those derived from TLR2[+]c-kit[+] cells being more abundant. Importantly, erythroid colonies from E7.5 TLR2[-]c-kit[+] progenitors expressed significantly higher ratio between embryonic and adult globins than their TLR2[+]c-kit[+] counterparts (Supplementary Fig. 2b). When scored at 12 days of culture, both c-kit[+] subsets produced erythroid and megakaryocytic colonies (E/Mk). However, a critical distinction between

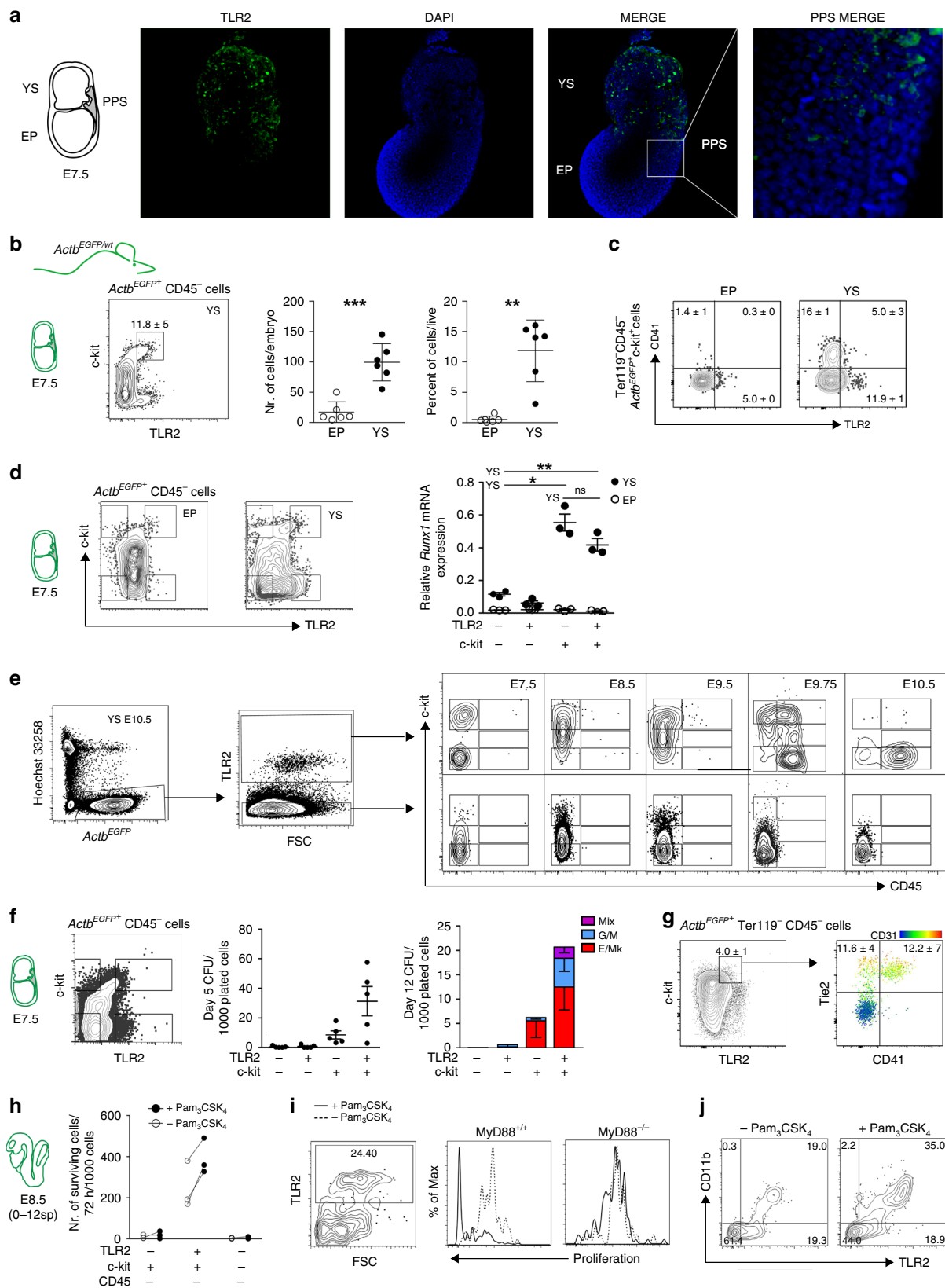

TLR2⁻c-kit⁺ and TLR2⁺c-kit⁺ progenitors was that only the latter gave rise to mixed colonies (Mix) (Fig. 1f; Supplementary Fig. 2b). In five independent experiments, apart from the occasional monopotent myeloid colonies (G/M), mixed colonies from TLR2⁻c-kit⁺ progenitors were never observed. This

principal distinction of EMPs, signified by TLR2 expression on their c-kit⁺ precursors, was functionally discernable despite the fact that a fraction of both TLR2⁺c-kit⁺, and to a lesser extent, TLR2⁻c-kit⁺ subsets, co-expressed Tie2, CD31 along with CD41 (Fig. 1g; Supplementary Fig 2c, middle row), suggesting their

**Fig. 1** Early YS-derived TLR2+ c-kit+ cells exhibit features of EMP precursors. **a** Immunofluorescence of E7.5 embryos revealed the presence of TLR2+ cells (green) predominantly in YS. Weaker TLR2 staining was also detected in PPS (white insert). Nuclei were stained with DAPI (blue). YS yolk sac, EP embryo proper, PPS posterior primitive streak. A representative image is shown ($n = 2$ independent experiments with at least four embryos per experiment). **b–j** $Actb^{EGFP/wt}$ embryos (see Supplementary Fig. 1b) were used to analyze cells of embryonic origin. **b** Quantification of $Actb^{EGFP}$+CD45⁻c-kit+TLR2+ cells in E7.5 EP and YS (mean ± SD; $n = 6$; **$p \le 0.01$, ***$p \le 0.001$; paired, two-tailed $t$ test). **c** Surface co-expression of TLR2 with CD41 determined on E7.5 $Actb^{EGFP}$+CD45⁻c-kit+ cells obtained from EP or YS. **d** $Runx1$ mRNA expression normalized to $Gapdh$ levels in four sorted subsets of E7.5 embryonic $Actb^{EGFP}$+CD45⁻ cells with combinatorial expression of c-kit and TLR2 (mean ± SD; $n = 3$; *$p \le 0.05$, **$p \le 0.01$; paired, two-tailed $t$ test). **e** E7.5- E10.5 YS $Actb^{EGFP}$+ cells of either TLR2+ or TLR2⁻ phenotype were analyzed for the expression of c-kit and CD45 by FCM. **f** E7.5 $Actb^{EGFP}$+Lin⁻ cells (Lin = CD3⁻Gr-1⁻CD11b⁻B220⁻Ter119⁻) were sorted based on their c-kit and TLR2 expression (see sorting strategy in Supplementary Fig. 2a), plated in a methylcellulose medium M3434 (StemCell Technologies) and assessed for their CFU potential (mean ± SEM; $n = 5$). **g** YS-derived E7.5 $Actb^{EGFP}$+Ter119⁻ CD45⁻c-kit+TLR2+ cells were analyzed for Tie2, CD41, and CD31 expression. See full gating strategy in Supplementary Fig. 2c. **h–j** Sorted E8.5 $Actb^{EGFP}$+ cells were plated on OP-9 stromal cells in the presence or absence of the TLR2 agonist Pam₃CSK₄ (1 μg/ml) and analyzed by FCM after 72 h ($n = 3$). **h** Survival analysis of sorted E8.5 TLR2+/⁻c-kit+/⁻CD45⁻cells. **i** The proliferation history of E8.5 TLR2+c-kit+CD45⁻ cells sorted from MyD88+/+ or MyD88⁻/⁻ embryos was assessed by the dilution of a proliferation dye in TLR2+ cells (see also Supplementary Fig. 2d). **j** $Actb^{EGFP}$+ cells recovered from cultures of $Actb^{EGFP}$+TLR2+ c-kit+ CD45⁻ cells stimulated, or unstimulated, with Pam₃CSK₄ were analyzed for CD11b and TLR2 expression. Source data are provided as a Source Data file

common hemogenic endothelial ancestry and hematopoietic fate. However, while t-SNE analysis revealed that these two subsets displayed a continuum of Tie2, CD31, and CD41 surface expression, it was the TLR2+c-kit+ subset which formed a tight cluster (red dots) with the highest expression of these markers (Supplementary Fig. 2c, bottom panel). Taken together, the expression of TFs, endothelial, and hematopoietic markers and the outcome from clonogenic assay link TLR2⁻c-kit+ progenitors to primitive erythro/megakaryopoiesis. In contrast, TLR2 expression on E7.5 c-kit+ YS progenitors accompanied by the high expression of Tie2, CD31 and CD41 predicates the acquisition of functional competence for multi-lineage EMP potential. Thus, TLR2 expression allows for the unequivocal distinction of emerging precursors of EMPs among coexisting progenitors of primitive erythropoiesis.

**TLR2 stimulation of TLR2+c-kit+ precursors enhances the production of myeloid cells**. When E8.5 $Actb^{EGFP}$+ cells of either TLR2+c-kit+, TLR2⁻c-kit+, or TLR2⁻c-kit⁻ phenotypes were co-cultured with OP-9 stroma in the presence or absence of the TLR2 agonist, Pam₃CSK₄, both TLR2⁻ populations failed to survive. However, ~39 and 24% of $Actb^{EGFP}$+ cells that were either stimulated or not stimulated with Pam₃CSK₄, respectively, were recovered from co-cultures with TLR2+c-kit+ progenitors (Fig. 1h). The increased survival after TLR2 stimulation correlated with a higher rate of proliferation that was fully dependent on MyD88, the adaptor protein which is required for TLR2 signaling (Fig. 1i; Supplementary Fig. 2d). In addition, Pam₃CSK₄-stimulated cells showed a more efficient myeloid differentiation rate measured by the expression of CD11b (Fig. 1j). This data, as well as the findings of a previous report, suggest that the capacity of activated TLRs to nudge the output of hematopoiesis toward myelo/granulopoiesis is inherent to all progenitors of definitive hematopoiesis, ontogenetically commencing with the emergence of TLR2+c-kit+ precursors of EMPs[29].

***Tlr2* locus is efficiently activated in erythro-myeloid progenitors**. To follow the fate of cells with an active *Tlr2* locus, we generated a $Tlr2^{Cre}$ mouse strain by BAC recombineering. In adult animals, *Tlr2* activation labeled all hematopoietic lineages and their progenitors with no significant bias (Supplementary Fig. 3). To determine the first ontogenetic time point of *Tlr2*-driven labeling, we looked at the emergence of $Tlr2^{Cre}$EYFP+ cells in E7.5–E10.5 embryos. Due to the expected delay caused by Cre-

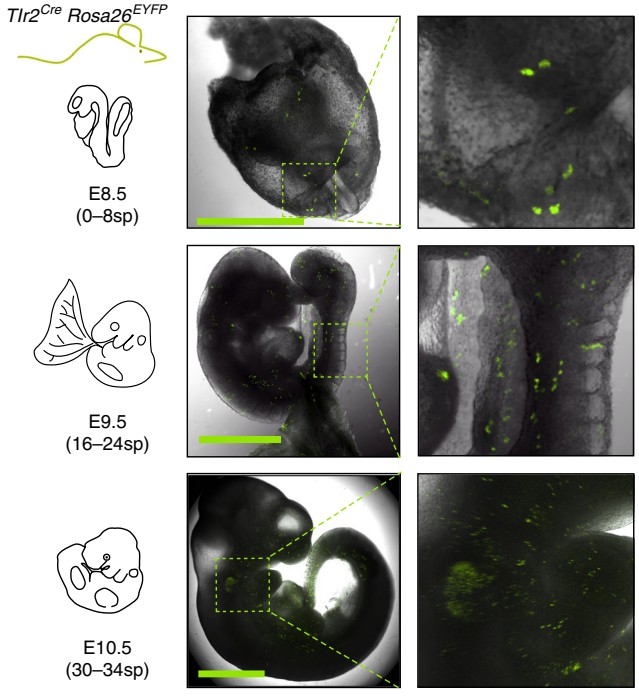

**Fig. 2** Lineage tracing shows early activation of *Tlr2* locus in hematopoietic progenitors. Spatial microscopic analysis of E8.5–E10.5 $Tlr2^{Cre}$EYFP+ cells; scale bar = 1 mm. Representative image is shown ($n = 3$ independent experiments with 3–6 embryos per experiment)

mediated recombination followed by EYFP labeling, the first $Tlr2^{Cre}$EYFP+ cells appeared at E8.5 (0–8sp) (Fig. 2).

If TLR2 expression marks the emergence of EMP precursors, then EMPs and their progeny with a sufficiently activated *Tlr2* promoter should be genetically labeled in the *Tlr2*-driven reporter mice. Thus, we first assessed the phenotype and frequency of labeled cells by combining Ter119, c-kit, CD41, and FcRγ surface markers to discriminate erythroid and myeloid precursors[6]. The analysis of $Tlr2^{Cre}$EYFP+ cells at E8.5, E9.5 YS, and EP revealed the presence of five distinct populations that represented the most abundant $Tlr2^{Cre}$EYFP+ subsets (Supplementary Fig. 4a). At E8.5, $Tlr2^{Cre}$EYFP+ cells accounted for an average 6.9% of the total cells, among which non-hematopoietic cells and primitive erythrocytes (EryP) each represented ~40%. Subsets of c-kit+FcRγ+ (MFp), c-kit+CD41+ (Mkp), and c-kit+CD41+FcRγ+ (EMPs) phenotypes

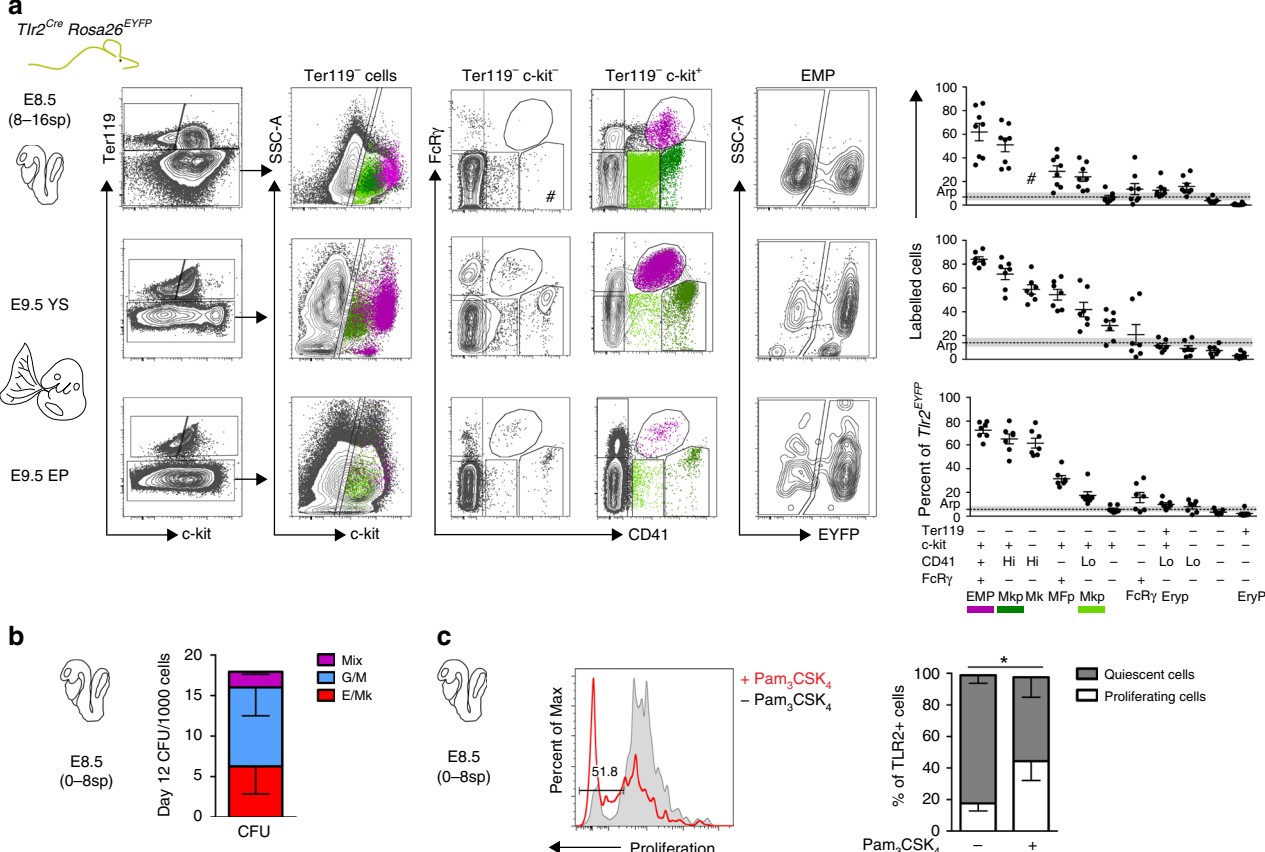

**Fig. 3** Lineage tracing shows activation of *Tlr2* locus predominantly in EMPs. **a** Embryonic hematopoietic precursors were analyzed by FCM for frequency of labeling in E8.5 and E9.5 *Tlr2^CreRosa26^EYFP* embryos (mean ± SEM; $n = 7$–9; arp, average recombination probability is equal to mean ± SEM (gray zone) labeling efficiency of all live cells; EMP, erythro-myeloid progenitor; Mkp, megakaryocyte progenitor; Mk, megakaryocyte; MFp, macrophage progenitor; FcRγ, FcRγ+ cells; Eryp, Ery progenitor; EryP, primitive erythrocyte; # insufficient cell count for statistical analysis). **b** Sorted E8.5 *Tlr2^Cre^*EYFP+Lin− cells (see Supplementary Fig. 4g) were plated in a methylcellulose medium and assessed for their differentiation potential at day 12 of culture (mean ± SEM; $n = 8$). **c** The proliferation history of sorted *Tlr2^Cre^*EYFP+Lin− cells cultured in the presence (red open histogram) or absence (gray closed histogram) of Pam₃CSK₄ (1 μg/ml) on ST-2 stroma was assessed by the dilution of proliferation dye in TLR2+ cells after 72 h (mean ± SEM; $n = 3$; *$p < 0.05$; paired, one-tailed $t$ test). Source data are provided as a Source Data file

collectively represented the remaining ~20% of *Tlr2^Cre^*EYFP+ cells (Supplementary Fig. 4a). While being at E8.5 the least frequent population among *Tlr2^Cre^*EYFP+ subsets, YS EMPs, which expressed the highest levels of c-kit, exhibited the highest labeling efficiency (~60%, Fig. 3a, top scatterplot) attesting to the preferential and robust activation of their *Tlr2* locus. Other populations, Mkp, MFp, and by E9.5 also CD41+ Mk, were also preferentially labeled well above the average recombination probability (arp) threshold. In contrast, EryPs were negligibly labeled (Fig. 3a, scatterplots). In the E9.5 YS, due to their expansion, EMPs accounted for 17–38% of all *Tlr2^Cre^*EYFP+ cells with even higher labeling efficiency (< 80%) (Supplementary Fig. 4a and Fig. 3a, middle scatterplots). Consistent with previous reports, at E9.5, only a fraction of TLR2+ EMPs were found in EP compared to YS (Supplementary Fig. 4b) confirming that these cells emerge in YS[4,35].

Interestingly, when maternally-derived FcRγ+ MFs (mMFs) were discounted, nearly one-half of the minute population of E8.5 FcRγ+ cells (eFcγ) was also *Tlr2*-labeled (Supplementary Fig. 4c). However, as these cells, in contrast to mMFs (Supplementary Fig. 1b, right panel), were virtually all CD45−CD11b− (Supplementary Fig. 4c) they could not be considered *bona-fide* MFs. Moreover, since they represented merely 0.3% of all *Tlr2*-labeled

cells, it is impossible at this junction to make any statement about their origin and affiliation to a certain hematopoietic wave and lineage. However, *Tlr2^Cre^*EYFP+ tissue-resident MFs were microscopically detectable in the E11.5 head and FL (Supplementary Fig. 4d). In addition, and consistent with the activation of *Tlr2* locus in the progenitors of EMPs, while E11.5 FL early erythroid progenitors (Early E)[36] were labeled almost as efficiently as FL and circulating EMPs, circulating and FL EryPs were labeled with only background and sub-background frequencies, respectively (Supplementary Fig. 4e, f).

Functionally, *Tlr2^Cre^*EYFP+Lin− cells isolated from E8.5 (0–8sp) embryos (Supplementary Fig. 4g) exhibited the potential to generate erythroid, myeloid, and mixed colonies (Fig. 3b). In addition, E8.5 *Tlr2^Cre^*EYFP+Lin− cells responded to TLR2 triggering by an enhanced proliferation rate (Fig. 3c). Thus, the expression of functional TLR2 accompanies the emergence of EMPs in vivo.

### E8.5 *Tlr2*-labeled progenitors contribute to embryonic hematopoiesis. To follow the developmental fate of cells with active *Tlr2* locus in a time controlled manner, we generated a *Tlr2^CreERT2^* mouse BAC strain (Supplementary Fig. 5a). In this model, *Cre* expression and the generation of EYFP+ cells was

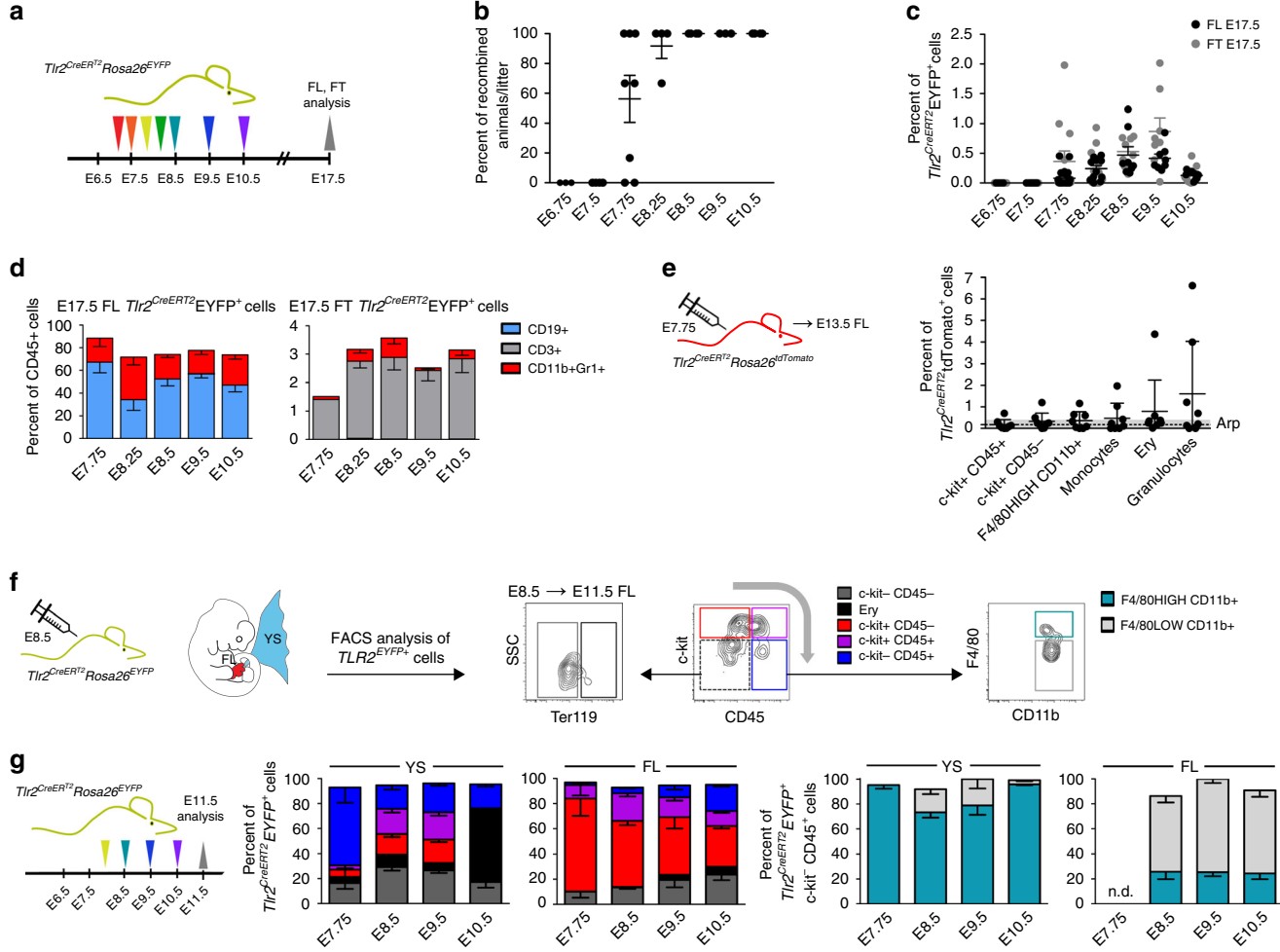

**Fig. 4** At E7.75, *Tlr2*-labeled progenitors contribute to embryonic and fetal hematopoiesis. **a** Experimental design of the fate mapping of cells with an active *Tlr2* locus. **b** The percentage of EYFP+ embryos among all *Tlr2^CreERT2Rosa26^EYFP* embryos in the same litter pulsed with one dose of 4-OHT from E6.75 to E10.5 was analyzed at E17.5 by FCM (mean ± SEM; n = 3–8 litters). **c** Percentage of *Tlr2^CreERT2*EYFP+ cells in E17.5 fetal liver (FL, black dots) and E17.5 fetal thymus (FT, gray dots) in embryos described in (**b**); (mean ± SEM; n = 6–21 embryos). **d** The hematopoietic fate of *Tlr2^CreERT2*EYFP+ cells pulsed with one dose of 4-OHT from E7.75 to E10.5 was determined in E17.5 FL and FT by FCM; (mean ± SEM; n = 4–8). **e** Labeling of hematopoietic populations (including granulocytes and monocytes) was determined in E13.5 FL of *Tlr2^CreERT2Rosa26^tdTomato* embryos pulsed with a single dose of 4-OHT at E7.75 (mean ± SEM; n = 8; arp = average recombination probability assessed in Ter119−c-kit−CD45− cells. **f** Gating strategy to identify the contribution of *Tlr2^CreERT2*EYFP+ cells to indicated hematopoietic populations in E11.5 YS and FL. **g** Fate tracing of E11.5 *Tlr2^CreERT2*EYFP+ cells pulsed with one dose of 4-OHT from E7.75 to E10.5 (mean ± SEM; n = 5–15) was analyzed by FCM according to (**f**). Source data are provided as a Source Data file

accomplished within 18 h upon 4-hydroxy tamoxifen (4-OHT) administration (Supplementary Fig. 5b). The first *Tlr2^CreER-T2Rosa26^EYFP* recombined embryos were found when 4-OHT was applied at E7.75 (Fig. 4a, b), resulting in the appearance of labeled cells in the FL and thymus (FT) (Fig. 4c). A single 4-OHT pulse at E7.75, E8.5, E9.5, or E10.5 marked hematopoietic progenitors that in E17.5 FL and FT gave rise to all main hematopoietic lineages (Fig. 4d; Supplementary Fig. 5c) with the highest labeling efficiencies at E8.5 and E9.5 (Fig. 4c). Moreover, consistent with the initiation of TLR2 expression in the precursors of EMPs at E7.5, when pulsed at E7.75, myeloid cells, including Ly6G+ granulocytes and Ly6C+ monocytes, were labeled in E13.5 *Tlr2^CreERT2Rosa26^tdTomato* FL (Fig. 5e; Supplementary Fig. 5d), supportive evidence of their early EMP origin. It is of note, that in the absence of 4-OHT, no labeling was observed (Supplementary Fig. 5e). This data confirm that the E7.75 embryo already contains a fully committed pool of multi-lineage progenitors of EMP and LMP.

We next investigated the fate of *Tlr2^CreERT2*EYFP+ cells. We pulsed *Tlr2^CreERT2Rosa26^EYFP* embryos from E7.75 to E10.5 with a single dose of 4-OHT and assessed the phenotype of *Tlr2^CreERT2*EYFP+ cells at E11.5 in YS and FL by flow cytometry (FCM) (Fig. 4f, g). When pulsed between E7.75 and E9.5, *Tlr2^CreERT2*EYFP+ cells at their c-kit+ stage colonized FL where they retained mostly a c-kit+ phenotype with only a fraction differentiating to c-kit+CD45+ cells. In YS, *Tlr2^CreERT2*EYFP+ cells differentiated to c-kit+CD45+ and CD45+ cells, a vast majority of which were F4/80^HIGHCD11b+ MFs. In contrast, a minute CD45+ myeloid population in FL was predominantly F4/80^LOWCD11b+. Importantly, the highest labeling efficiency of E11.5 FL cells was achieved upon labeling from E7.5 to E8.5 (Supplementary Fig. 5f). These results imply that the wave of E7.5–8.5 *Tlr2^CreERT2*EYFP+ EMP precursors contributed the most to the pool of hematopoietic cells differentiating in the FL.

When tracing the fate of cells labeled by *Tlr2* at E8.5 at later stages of development (E12.5–E15.5; Supplementary Fig. 5g), we

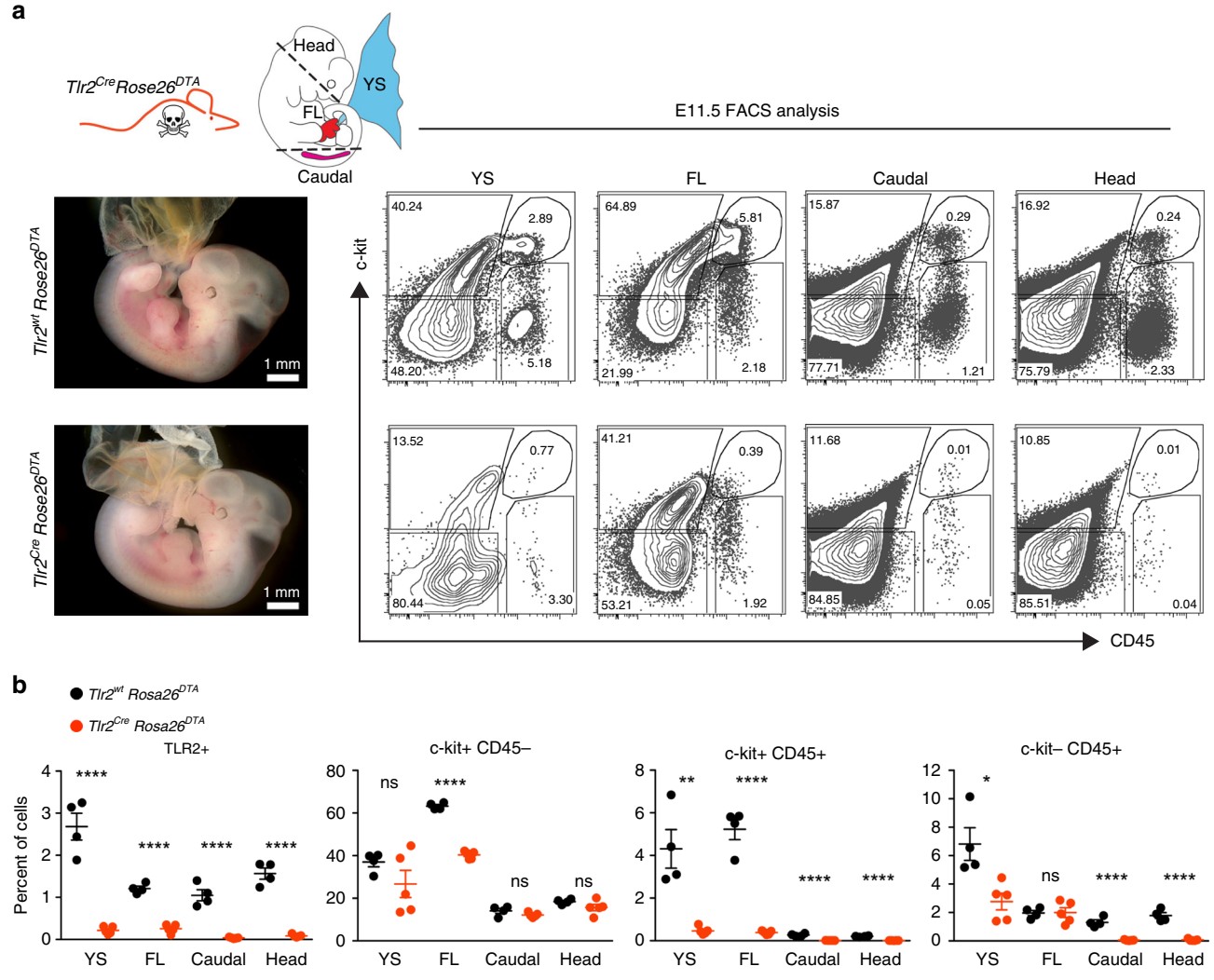

**Fig. 5** *Tlr2*-driven production of DTA efficiently deletes hematopoietic cells in E11.5 embryos. **a** Representative images of E11.5 *Tlr2wtRosa26DTA* and *Tlr2CreRosa26DTA* embryos (n = 4 independent experiments with 3–8 embryos of each genotype per experiment). FCM analysis (**a**) and its quantification (**b**) of TLR2+, c-kit+CD45−, c-kit+CD45+, and c-kit−CD45+ cell subsets in the YS, FL, caudal part, and head of dissected E11.5 *Tlr2wtRosa26DTA* (upper panel) and *Tlr2CreRosa26DTA* (lower panel) embryos (mean ± SEM; n = 4-5; ****p ≤ 0.0001, *p < 0.05; paired, two-tailed t test). Scale bar = 1 mm. Source data are provided as a Source Data file

observed that c-kit+ cells gradually disappeared from YS as the number of myeloid and erythroid cells increased. Starting from E12.5, the number of labeled erythroid cells in FL increased at the expense of c-kit+ as well as c-kit+CD45+ progenitors, attesting for the relocalization of the main site of erythropoiesis from YS to FL.

***Tlr2*-driven production of diphtheria toxin efficiently deletes EMPs**. To determine the phenotype of embryos that lack TLR2+ cells, we employed a *Rosa26DTA* reporter strain whereby the activation of the diphtheria toxin (DTA) module causes cell death (Fig. 5a). Depletion of TLR2+ cells from E11.5 *Tlr2CreRosa26DTA* embryos was highly efficient (Fig. 5a, b, left scatterplot; Supplementary Fig. 6a) and caused lethality before E13.5 (Supplementary Fig. 6b). At E11.5, the absence of TLR2+ cells translated into a significant decrease of myeloid cells as well as their progenitors in the YS, head, and caudal part of the embryo (Fig. 5a, b). By E11.5, c-kit+CD45+ progenitors had vanished in all compartments. Consistent with *Tlr2* activity in tissue MFs, microglia were also efficiently depleted (Supplementary Fig. 6c). While the depletion of TLR2+ cells did not affect the counts of EryPs in

E12.5 peripheral blood, it caused severe defects in FL definitive hematopoiesis (Supplementary Fig. 6d, e). E12.5 *Tlr2CreRosa26DTA* livers were pale and had reduced cellularity as compared with livers of wt animals. In addition, EMPs and MFs were virtually absent and the counts of Early E, as well as erythroid cells were severely decreased (Supplementary Fig. 6e). Thus, while primitive erythropoiesis remained intact, EMP-derived erythropoiesis was severely affected upon depletion of TLR2+ cells. It is of note, that even though *Tlr2* locus is activated also in some non-hematopoietic cells, their numbers were not affected by *Tlr2* driven depletion, strongly advocating that lethality was caused by aberrant EMP-dependent hematopoiesis. Thus, the *Tlr2* locus is activated in the earliest YS-derived c-kit+ precursors of EMPs which in *Tlr2CreRosa26DTA* embryos, due to DTA toxicity, largely fail to mature to the erythro-myeloid progenitor stage and beyond.

***Tlr2* locus is active in hematopoietic clusters emerging from the mouse aorta**. Analysis of adult *Tlr2CreRosa26EYFP* mice BM showed that the *Tlr2* locus is active in phenotypical LT-HSC (Supplementary Fig. 3d). To determine whether it is also active in

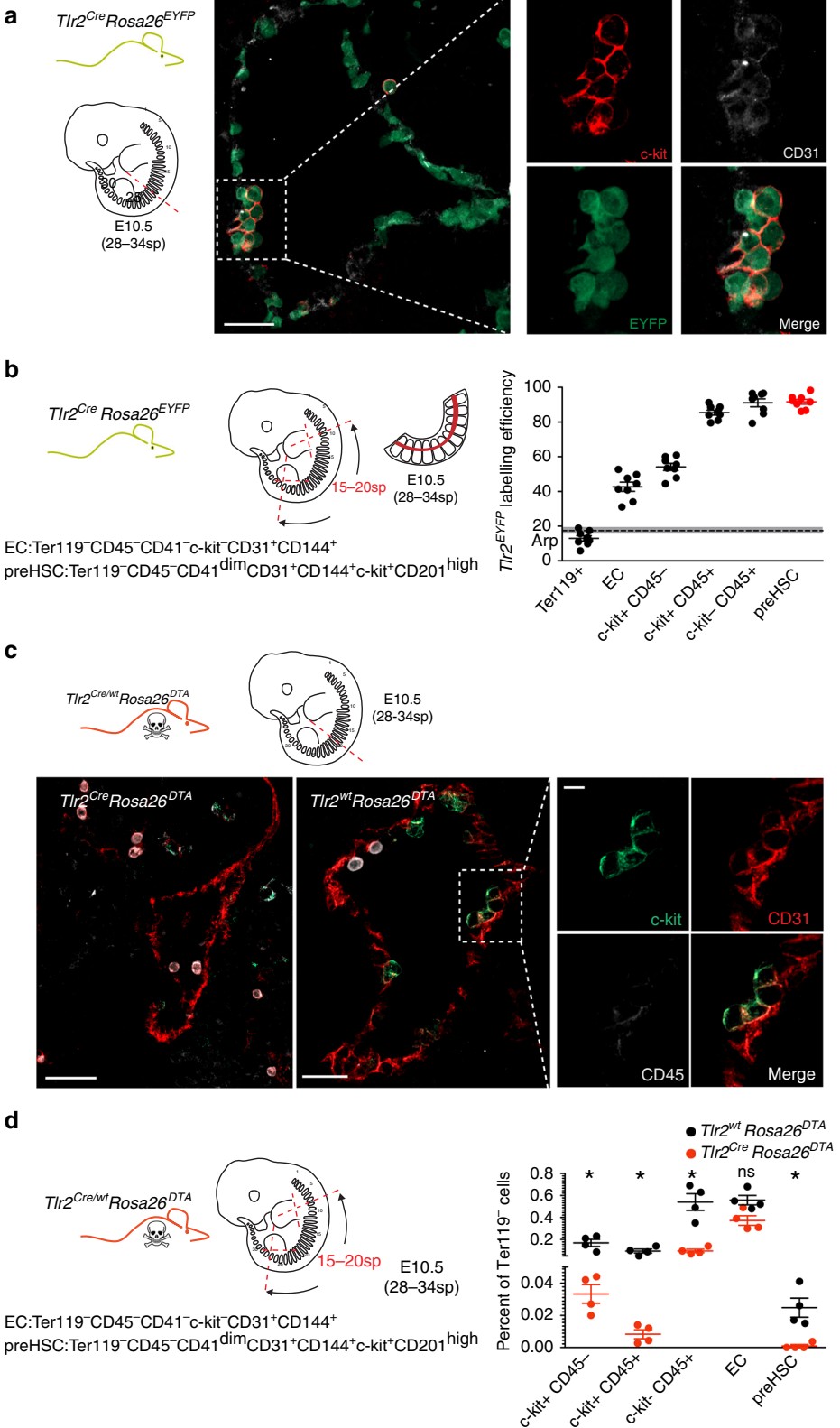

HSCs during embryonic development, we imaged the aortic sections of E10.5 $Tlr2^{Cre}Rosa26^{EYFP}$ embryos (28–34sp) where intra-aortic hematopoietic clusters (IAHCs) emerged from the CD31$^+$ endothelium[10,37]. In addition to endothelial cells at the lining of the aorta, IAHCs were also labeled (Fig. 6a). Moreover,

in the aortic parts of these embryos, pre-HSCs[38] showed the most efficient labeling among all populations tested (98% ± 4.0 SD) as opposed to endothelial cells which were only partially labeled (40% ± 7.5 SD) (Fig. 6b; Supplementary Fig. 7a). This suggests that activation of the $Tlr2$ locus in blood cells occurred at the time

**Fig. 6** E10.5 aortic pre-HSC activate their *Tlr2* locus. **a** Cryosections of E10.5 (28–34sp) dorsal aortae with intra-aortic hematopoietic clusters (IAHCs) were analyzed from *Tlr2^CreRosa26^EYFP* embryos by staining for EYFP (green), c-kit (red) and CD31 (white). A representative image is shown (*n* = 5 independent experiments with 1–2 embryos per experiment). **b** Aortic regions were dissected from E10.5 *Tlr2^CreRosa26^EYFP* embryos (28–34sp), and hematopoietic populations as well as endothelial cells were analyzed for their labeling efficiency by FCM. The gating strategy for each indicated subset adopted from ref., [38] is shown in Supplementary Fig 7a. **c** Sections of E10.5 (28–34sp) aortae were taken from *Tlr2^CreRosa26^DTA* (left panel) and *Tlr2^wtRosa26^DTA* (middle and right panels) embryos and stained for c-kit (green), CD31 (red), and CD45 (white) to identify IAHCs. Scale bar represents 50 μm. A representative image is shown (*n* = 5 independent experiments with 1–2 embryos of each genotype per experiment). **d** Aortic regions were disected from E10.5 *Tlr2^wtRosa26^DTA* (black dots) and *Tlr2^CreRosa26^DTA* (red dots) embryos (28–34sp). Hematopoietic populations as well as endothelial cells were analyzed for their frequencies by FCM (mean ± SEM; *n* = 4; *$p < 0.05$; paired, two-tailed *t* test). Gating strategy is shown in Supplementary Fig. 7b. Source data are provided as a Source Data file

of HSCs specification. In addition, the emergence of IAHCs was disrupted in *Tlr2^CreRosa26^DTA* embryos (Fig. 6c) where pre-HSCs were absent (Fig. 6d; Supplementary 7b).

**E8.5 HSC progenitors with an active *Tlr2* locus contribute to adult hematopoiesis.** To address if early *Tlr2^CreERT2*EYFP+ progenitors contribute to adult hematopoiesis, we pulsed *Tlr2^CreERT2Rosa26^EYFP* embryos from E6.75 to E10.5 with a single dose of 4-OHT and monitored *Tlr2^CreERT2*EYFP+ cells in the peripheral blood (PB) of animals until 16 weeks of age (Fig. 7a). While the first time point of PB labeling of all three main hematopoietic lineages occurred at ~E7.75–8.0, it persisted for only the first 4 weeks of life, thus targeting EMPs and LMPs (Fig. 7a). The highest efficiency of labeling was achieved when pulsed at E8.5 (Fig. 7a), whereby *Tlr2^CreERT2*EYFP+ cells contributed to all three main hematopoietic lineages in all hematopoietic tissues tested at 16 weeks of age (Fig. 7b; Supplementary Fig. 8a). When the E15.5 FL of these mice were analyzed, ST-HSC, multipotent progenitors (MPP), and LT-HSCs were labeled at the highest frequencies (Fig. 7c). Moreover, E8.5 *Tlr2*-labeled Lin− BM cells (*n* = 8) engrafted primary, lethally irradiated recipients for more than 16 weeks (Fig. 7d; Supplementary Fig. 8b) Then, lineage-depleted *Tlr2^CreERT2*EYFP+ sorted cells isolated from BM of primary recipients 16-weeks after transplantation engrafted secondary adult lethally irradiated recipients giving rise to all three hematopoietic lineages (Fig. 7d; Supplementary Fig. 8c) demonstrating their ability to self-renew. These data imply that *Tlr2^CreERT2*EYFP+cells that give rise to HSCs activated their *Tlr2* locus as early as E8.5. This stage so far represents one of the earliest known events in the HSC specification process.

## Discussion
Our study has established that the expression of TLR2 on c-kit+ cells allows for the discrimination of emerging multi-lineage precursors of EMPs from c-kit+ progenitors of the primitive erythroid wave, both of which are present in E7.5 embryos. Notably, phenotypic analysis and clonogenic assay demonstrated that only TLR2+c-kit+ but not TLR2−c-kit+ cells are able to mature to CD45+ hematopoietic cells and produce mixed colonies. The data from *Tlr2^CreERT2Rosa26^EYFP* and *Tlr2^CreRosa26^EYFP* embryos confirmed that the *Tlr2* locus in hematopoietic precursors is activated at ~E7.5, which is followed at E8.5 by the emergence of the first EYFP-labeled c-kit+ cells exhibiting essential EMP characteristics. Importantly, genetic ablation of these progenitors with active *Tlr2* locus left the primitive erythropoiesis intact. This data is in agreement with previously reported labeling of EMPs by 4-OHT at E7.75 in *Csf1r^MeriCreMer* embryos[22]. Indeed, *Tlr2* expressing cells were enriched in the hematopoietic progenitor cluster identified by single cell profiling of E7.75 cells[39]. Consistent with the co-expression of TLR2 on c-kit+ progenitors shown in this study, the emergence of the earliest pre-HSCs at E8.5 can be visualized by

lineage tracing using both *Tlr2*- and *c-Kit*-based reporter systems[24].

The co-expression of TLR2 and c-kit markers as the earliest signature of emerging EMPs precursors aids to clarify the sequence and developmental relationships among early hematopoietic waves. Since it has been assumed that the primitive and EMP waves emerge successively at E7.5 and E8.5, respectively[40], preferential labeling of brain microglia over other tissue resident MFs at E7.5 using *Runx1*, *Tie2*, or *c-Kit* drivers[22–24], has led some to propose that they originated from the primitive wave. Our data supports an alternative view that, due to their appearance at E7.5, c-kit+TLR2+ precursors of EMPs could represent the main source of progenitors for brain microglia, fetal, and adult tissue-resident MFs as well as fetal definitive erythrocytes[4,41]. The brain vasculature develops early, hence it becomes a primary sink of migrating precursors of EMPs[42]. In line with previous reports, we also showed that EMPs, the precursors of which expressed TLR2 between E7.75 and E8.5, were the main source of cells seeding FL, where they expanded and differentiated mostly to monocyte-like F4/80^LOWCD11b+ myeloid cells, whereas beyond FL they preferentially produced F4/80^HIGHCD11b+ tissue resident MFs[6,22]. Thus, while our data advocates a shared endothelial origin and an overlapping expression pattern of *Runx1*, CD41, Tie2, and CD31 between E7.5 progenitors of the primitive and EMP waves, it is the acquisition of TLR2 and the enhanced expression of CD41, Tie2 and CD31 on the background of c-kit that marks the initiation of the ontogenetic program which endows early progenitors with the acquisition of multi-lineage erythro-myeloid potential. Endogenous labeling of cells with an active *Tlr2* locus showed that EryPs were labeled only at a very low efficiency, hardly overcoming the background labeling level, suggesting that the activation of the *Tlr2* locus occurs predominantly and most robustly in emerging EMPs and their progeny. Thus, the contribution of *Tlr2*-labeled EryPs to their total pool is negligible as is the effect of their depletion on early embryonic erythropoiesis. At E8.5, ~60–90% of EMPs were labeled by *Tlr2* activation, followed by progenitors of MFs, Mks, and Eryps.

Functionally analogous to the situation in adult BM[29], stimulation of E8.5 TLR2+c-kit+ progenitors and E8.5 *Tlr2^Cre*EYFP+c-kit+Lin− cells by the TLR2 ligand enhanced their proliferation rate as well as differentiation to myeloid cells in a *MyD88*-dependent manner. However, as TLR2+ embryonic progenitors were still able to proliferate and differentiate into myeloid cells regardless of ligand stimulation, TLR2-generated signals seemed to be dispensable for normal embryonic myelopoiesis under noninflammatory conditions. Presently, no evidence for the impaired development of myeloid cells at steady-state in *Tlr2^−/−* or *MyD88^−/−* mice has been reported[43,44]. Thus, the early activation of the *Tlr2* locus and its dispensability for the process of embryonic development, makes *Tlr2* regulatory elements a suitable target for the generation of experimental models for fate mapping of the earliest steps in embryonic hematopoiesis.

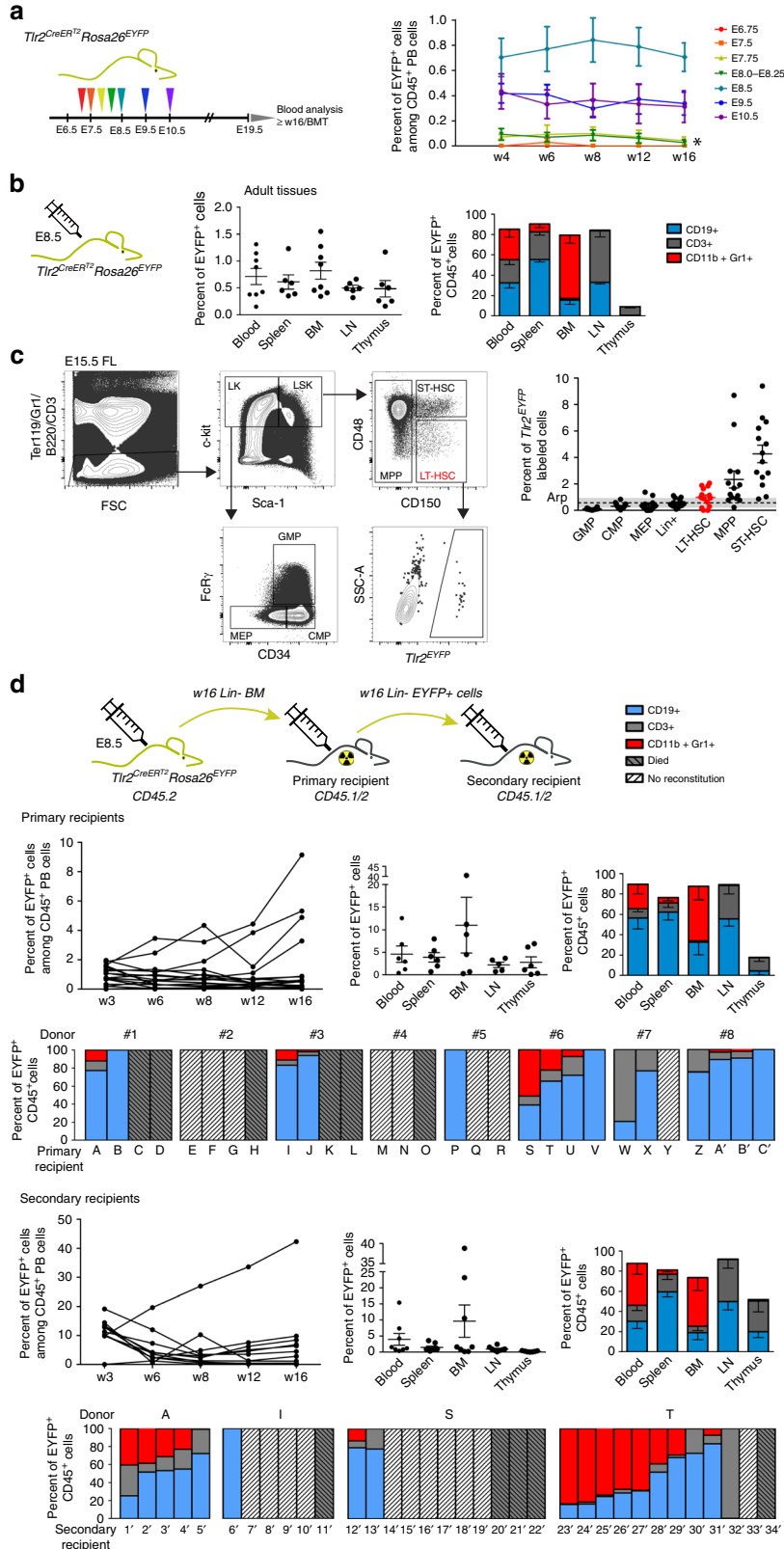

At E8.5, in *Tlr2^CreERT2^Rosa26^EYFP^* embryos, in addition to EMPs, HSC progenitors were labeled at the highest efficiency, which correlates with the expression of *Tlrs* on hematopoietic cluster cells in *Ly6a^GFP^* transgenic animals[32]. EMPs and HSCs originate from distinct populations of endothelial cells[17], yet the onset of their *Tlr2* promoter activity at E8.5 seems to be

ontogenetically synchronized. The indication that the activation of the inflammatory expression program occurs during EHT[32,45], together with our finding that TLR2 stimulation of the earliest TLR2+c-kit+ YS precursors as well as BM-derived HSCs[29] augmented their myeloid fate, suggests that the mechanism of TLR-driven hematopoiesis under inflammatory conditions is not only

**Fig. 7** Progenitors with active *Tlr2* locus labeled at E8.5 contribute to adult hematopoiesis. **a** *Tlr2*^CreERT2*Rosa26*^EYFP embryos pulsed with a single dose of 4-OHT between E6.75-E10.5 were monitored postpartum for the presence of *Tlr2*^CreERT2^EYFP+ cells in their peripheral blood (PB) up to 16 weeks of age (mean ± SEM; *n* = 9–25; * indicates short-term hematopoietic potential which persisted for only the first 4 weeks of life. **b** Percentage of *Tlr2*^CreERT2^EYFP+ cells, labeled at E8.5, and their contribution to the main CD45+ lineages in different hematopoietic organs is shown (mean ± SEM; *n* = 5–7; see gating strategy in Supplementary Fig. 8a). **c** *Tlr2*^CreERT2*Rosa26*^EYFP embryos were pulsed with a single dose of 4-OHT at E8.5, and labeling efficiency of indicated hematopoietic populations was analyzed in E15.5 FL by FCM. LT-HSC gate is shown as an example. **d** *Tlr2*^CreERT2*Rosa26*^EYFP embryos were pulsed with a single dose of 4-OHT at E8.5. At 16 weeks of age, Lin-depleted BM (CD45.2) (w16 Lin⁻ BM; see Supplementary Fig. 8b) was transferred to lethally irradiated primary recipients (CD45.1/2) along with support BM cells (0.5 × 10⁶ CD45.1), and the level of *Tlr2*^CreERT2^EYFP+ cells was monitored in PB of primary recipients until week 16 (upper left panel). The percentage of *Tlr2*^CreERT2^EYFP+ cells, their organ distribution and fate in primary recipients is shown (mean ± SEM; *n* = 6). Bar graphs show the reconstitution success and fate of *Tlr2*^CreERT2^EYFP+ cells in primary recipients (A–C′) of BM from eight individual donors (#1–8). Sorted *Tlr2*^CreERT2^EYFP+ cells (see Supplementary Fig. 8c) from Lin-depleted BM of 16-week old primary recipients were transferred to secondary, lethally irradiated recipients (CD45.1/2) along with support BM cells (0.5 × 10⁶, CD45.1) and the frequency of *Tlr2*^CreERT2^EYFP+ cells in PB was monitored until week 16 (the outcome of BMT from one primary recipient (donor T) is shown (secondary recipients, upper row, left panel). The percentage of *Tlr2*^CreERT2^EYFP+ cells, their organ distribution and fate in secondary recipients is shown (mean ± SEM; *n* = 8). Bar graphs on the bottom indicate the reconstitution success and fate of *Tlr2*^CreERT2^EYFP+ cells in secondary recipients (1′−34′) of BM from four primary recipients. Source data are provided as a Source Data file

operational in adult but also in embryonic hematopoietic progenitors. Indeed, a number of studies has reported that proinflammatory cytokines are important players in HSCs specification in the mouse and zebrafish[32,46,47]. In this scenario, TLR expression represents one of the oldest, evolutionary conserved and ontogenetically synchronized genetic programs activated during hematopoietic specification.

Taken together, the *Tlr* expression program is not only operational in embryonic and adult hematopoietic progenitors, endothelial cells, and MFs regardless of their origin[33] but it is initiated in the earliest YS-derived definitive hematopoietic progenitors as well as ancestors of HSCs.

## Methods

**Animals**. CD1 and C57Bl/6J mice were maintained at the animal facility of the Institute of Molecular Genetics in Prague (IMG). pCAGEGFPMos3 mice expressing EGFP under the control of the *Actb* promoter, here referred to as *Actb*^EGFP mice were generated and provided by Petr Svoboda[48]. *MyD88*⁻/⁻ mice (B6.129P2 (SJL)-*Myd88*^tm1.1Defr/J) and *Rosa*^DTA (B6.129P2-*Gt(ROSA)26Sor*^tm1(DTA)Lky/J) reporter mice were purchased from The Jackson Laboratory. *Rosa26*^tdTomato (B6;129S6-Gt(ROSA)26Sortm14(CAG-tdTomato) Hze/J) and *Rosa26*^EYFP (B6.129 × 1-Gt(ROSA)26Sortm1(EYFP)Cos/J) reporter mice were provided by V. Korinek. *Tlr2*^Cre and *Tlr2*^CreERT2 transgenic mice were generated by inserting *Cre* and *Cre*^ERT2 cassettes, respectively, under the control of the *Tlr2* promoter by BAC recombineering[49]. Murine BAC clone RP23-455F23 encompassing *Tlr2* gene with extensive upstream (46 kb) and downstream (123 kb) sequences was obtained from the BAC resource depository at CHORI. By homologous recombination, the open-reading frame of *Tlr2* exon 3 was substituted with *Cre* and *Cre*^ERT2 cDNA, respectively. BAC DNA for pronuclear injection was prepared using the QIAGEN Large-Construct Kit and analyzed by pulsed-field gel electrophoresis. The pronuclear injection of the construct (1 ng/µl) into mouse zygotes was carried out at the Transgenic Unit of IMG. Founders were identified by PCR amplification of tail DNA with specific primers for *Tlr2*^Cre transgenic and wt alleles: Tlr2F (common) 5′-AGCCATAGGCCACATCTAGT-3′, Tlr2CreR 5′-GTATGCTCAGAAAACGC CTG-3′ (470b), and Tlr2wtR 5′-AAAAAGCGATGTTACCCC-3′ (784b) and backcrossed to C57BL/6 mice. All experiments included littermate controls with the minimum sample size of three animals. Embryonic development was estimated by considering the day of vaginal plug formation as embryonic day 0.5 (E0.5) and staged using standard criteria[50]. To obtain E7.5 embryos, we used E7.5 time pregnant females and strictly selected only those embryos that were at the neural plate stage (NPS) with no visible signs of alantoic bud or structure, with an apparent cranial limit furrow and pointy node (i.e. E7.25–7.5, on rare occassion up to 7.75). All embryos, which were older than those without the "alantois bud stage" of NPS, were excluded from CFU assays experiments. In various other experiments, somite pairs were counted to determine embryo age more accurately. All experiments were approved by the ethical committee of the IMG.

**Cell suspension preparation**. Time pregnant females were killed by cervical dislocation, and embryos were dissected from uteri. E7.5–8.5 embryos were carefully stripped of maternal decidua, Reichart's membrane and ectoplacental cone. For flow cytometry (FCM) and qPCR analyses and cell sorting, cells from E7.5-E8.5 dissected embryos were pooled from one litter. E9.5–12.5 embryos were separated from extraembryonic membranes (YS and amnion), and single embryos were analyzed unless stated otherwise. Embryos were washed in Hank's balanced saline

solution (HBSS) and dissociated using 1 mg/ml dispase (Invitrogen) in HBSS for 10 min at 37 ℃ with occasional gentle pipetting. The reaction was stopped by washing in 2% FCS in HBSS. Embryonic suspensions were then passed through a 50-µm cell strainer. Peripheral blood of adult animals was collected from the facial vein into PBS EDTA solution, and red blood cells were lysed in ACK solution.

**Flow cytometry and cell sorting**. After Fc receptor blocking (in experiments where FcRγ was not stained) by rat anti mouse FcRγ antibody (2.4G2; Biolegend), single-cell suspensions were stained with conjugated monoclonal antibodies for 30 min on ice. Where appropriate, cells were further incubated with streptavidin conjugates for 20 min. The full list of antibodies and secondary reagents used can be found in the reporting summary. Fluorescence data were acquired using LSRII flow cytometer (BD Biosciences). FCM analysis was performed using FlowJo software (FlowJo, LCC). Cell debris and dead cells were excluded from the analysis based on scatter signals and viability dye fluorescence (Hoechst 33258 (Sigma-Aldrich), Sytox-blue, Fixable viability dye eFluor780 (Thermo Fisher Scientific)). Cell sorting was performed with an Influx cell sorter (BD Biosciences). For t-SNE and hierarchical clustering analysis, raw or pre-gated FCS3.0 files were exported from acquisition software and imported into R environment where all subsequent analyses were carried out[51].

**t-SNE**. For dimensionality reduction and subsequent population choices across all measured parameters, we used Barnes-Hut implementation of t-Distributed Stochastic Neighbour Embedding (t-SNE)[52]. In Fig. S1H, a tSNE map was built using a tSNE plugin in FlowJo software (FlowJo, LCC).

**Hierarchical clustering analysis for flow cytometry**. Agglomerative unsupervised Hierarchical clustering analysis (HCA) was performed using Mahalanobis distance measure and Mahalanobis-based custom linkage[53]. From the resulting hierarchy, the clusters (cell populations) were selected based on dendrogram topology and a matrix of parameters scatterplots.

**Gene expression analysis**. The total RNA from whole embryos or sorted cells was isolated using a RNeasy Plus Micro Kit (Qiagen) and was reverse transcribed using Premium RevertAid (Fermentas) and random hexamers (Fermentas). Quantitative RT-PCR (qPCR) was performed using the LightCycler 480 SYBR Green I Master mix on a LightCycler 480 instrument (Roche). Each sample was tested in triplicate. The relative amounts of mRNA were calculated using LightCycler 480 1.5 software with *Casc3* or *Gapdh* mRNA levels as a reference gene. Primers were designed using UPL software (Roche). Intron-spanning assays were used when possible. Primer efficiencies were calculated using LightCycler 480 1.5 software. Primer sequences are listed in Supplementary Table 1. Data were analyzed using Prism 5.03 software (GraphPad). The expression level of different *Tlrs* and their adaptors highlighted in Supplementary Fig. 1a precluded the effect of primer efficiency factor.

**Tracking cells of maternal and embryonic origin**. To follow the cells of embryonic origin, we crossed wt CD1 females with *Actb*^EGFP males. Only EGFP+ embryos were included in subsequent analysis. Due to paternal inheritance, all *Actb*^EGFP cells were considered to be of embryonic origin (Supplementary Fig. 1b, left panel). To follow maternal cells, *Actb*^EGFP/wt females were crossed with wt CD1 males. Maternally derived *Actb*^EGFP cells were analyzed in wt embryos that did not inherit the maternal EGFP allele (~50%) (Supplementary Fig. 1b, right panel).

**In vitro assays**. Distinct subpopulations of E7.5 or E8.5 embryonic $Actb^{EGFP}+$ cells were sorted based on their TLR2, c-kit, and CD45 surface expression, plated on a semi-confluent layer of OP-9 cells[54] (gift from J.C. Zuniga-Pflucker) or ST-2 cells[55] (gift from L. Klein) and cultured in RPMI containing 5% FCS (Sigma-Aldrich). E7.5 sorted cells were supplemented with recombinant cytokines IL-3 (1 ng/ml), SCF (50 ng/ml) GM-CSF (3 ng/ml), and M-CSF (10 ng/ml) (Biolegend). A thousand cells from each E8.5 sorted $Actb^{EGFP}+$ population labeled with Cell Proliferation Dye eFluor® 670 (eBioscience) were co-cultured with OP-9 cells in the presence or absence of 1 µg/ml Pam₃CSK₄ (Invivogen) for 72 h. The total number of surviving $Actb^{EGFP}+$ cells were normalized to 10,000 events obtained by FCM analysis. E8.5 $Tlr2^{Cre}EYFP+$ Lin⁻ cells were sorted, stained with Cell Proliferation Dye eFluor® 670 (eBioscience) and cultured in the presence or absence of Pam₃CSK₄ (1 µg/ml) on ST-2 stroma. The proliferation history of embryonic cells was determined by the dilution of Cell Proliferation Dye® eFluor 670 according to the manufacturer's instructions.

**Colony-forming cell assay**. Sorted E7.5 or E8.5 cells were plated on methylcel-lulose medium M3434 GF (Stem Cell Technologies) according to the manu-facturer's instructions. Cultures were maintained at 37 °C in humidified air with 5% $CO_2$. Hematopoietic colonies were scored at days 5 and 12. For EryP clonogenic assays, M3434 was supplemented with 3U/ml erythropoietin and 10% FCS. Microscopic images were acquired using Nikon Diaphot 300 equipped with 10 × / 0.25 or Plan 20 × /0.4 objectives.

**Single-colony qPCR**. Individual erythroid colonies were imaged and individually transferred to 5 µl of lysis buffer (0.1 % BSA in RNAse free water supplemented with RNAse inhibitors) and instantly frozen on dry ice. Cell lysate was then used for reverse transcription (Superscript III RT, Thermo Fisher Scientific). qPCR was performed using the LightCycler 480 SYBR Green I Master mix in a LightCycler 480 instrument (Roche).

**Whole embryo ex vivo imaging**. $Tlr2^{Cre}EYFP+$ cells in E8.5-E10.5 $Rosa26^{EYFP}Tlr2^{Cre}$ embryos were imaged with a Nikon AZ-100 confocal micro-scope equipped with a ×4 objective. Images of E7.5 $Actb^{EGFP}$ embryos and E11.5 $Tlr2^{wt}Rosa26^{DTA}$ and $Tlr2^{Cre}Rosa26^{DTA}$ embryos were acquired using a Olympus SZX9 StereoZoom microscope equipped with a DF PLAPO 1xPF objective and a DP72 digital camera.

**Whole mount embryo immunohistochemistry and imaging**. E7.5 embryos were dissected from decidua, washed several times in ice-cold PBS, fixed overnight in PHEM fixative (80 mM PIPES, 5 mM EGTA, 1 mM $MgCl_2$, 25 mM HEPES at pH of 7.2, 3.7% formaldehyde, purified 0.1% Triton X-100) at 4 °C, and then rinsed three times in PBS. Embryos were blocked in PBS supplemented with 10% goat serum for 1 h. The primary antibody, purified rat anti-mouse TLR2 (clone 6C2, eBioscience) was added at a final concentration of 5 µg/ml for overnight at 4 °C and developed with the secondary goat anti-rat IgG labeled with Alexa 488. The isotype control antibody produced no specific staining. Embryos were then embedded with Vectashield mounting medium with 4′,6-diamino-2-phentlindole (DAPI) (Vector Laboratories) and prepared for scanning. Signals were visualized and digital images were obtained using a Zeiss LSM 780 equipped with two photon, argon and helium–neon lasers. For 3D image, individual confocal planes (25–30 planes in 1–2 µm intervals) were pro-jected to generate a single stacked 3D-reconstructed image using Imaris 7.3 (Bitplane).

**Preparation of aortic regions for imaging and FCM analysis**. Caudal parts of E10.5 embryos including somites were cut between forelimbs and hindlimbs. These parts and the aortae were gently flushed with HBSS using a syringe to remove blood cells. Aortic regions were either washed and fixed in 3.7% PFA in PBS for subsequent imaging or digested with dispase for FCM analysis.

**Immunofluorescence of embryonic sections and imaging**. Embryos were fixed with 3.7% PFA in PBS overnight at 4 °C, washed in PBS, transferred to 30% sucrose and mounted in OCT for freezing and sectioning. Sections (8–10-µm thick) were postfixed in 3.7% PFA in PBS, permeabilized in methanol for 10 min at −20 ºC (except for phalloidin staining), then blocked in 5% BSA in 1% BSA 0.1% Triton-X100 (PBT) and stained with antibodies in 1% BSA PBT. The complete list of antibodies and secondary reagents used can be found in the NR reporting sum-mary. The coverslips were mounted using Vectashield containing DAPI (Vector Laboratories). Sections were imaged with Leica DM6000 epifluorescence micro-scope equipped with HCX PL APO 10.0 × 0.40, HCX PL APO 20 × 0.7 and HCX PL APO 40 × 0.75 objectives or with a Dragonfly 503 spinning disc confocal microscope and/or equipped with a HC PL APO 20 × / 0.75 IMM objective.

**Continuous labeling of Tlr2 progenitors**. For fate-mapping analysis of Tlr2 precursors, $Rosa26^{EYFP}$ or $Rosa26^{tdTomato}$ females were crossed with $Tlr2^{Cre}$ males. The indicated tissues from embryos and adult F1 mice were analyzed by FCM and imunofluorescence.

**Pulse labeling of Tlr2 progenitors**. For genetic cell labeling of embryonic cells, we crossed $Rosa26^{EYFP}$ reporter mice with tamoxifen-inducible $Tlr2^{CreERT2}$ mice. Recombination was induced by a single i.p. injection of 1.5 mg 4-hydroxytamoxifen (4-OHT; H6278; Sigma) into pregnant females from E6.75 to E10.5. When 4-OHT was administered after E9.5, the delivery was assisted by C-section at E20.5 and pups were transferred to foster mothers.

**Embryonic lethality assay**. $Rosa26^{DTA/DTA}$ females were crossed with $Tlr2^{Cre/wt}$ males to obtain litters with equal numbers of wt ($Tlr2^{wt/wt}Rosa26^{DTA/wt}$) and tg ($Tlr2^{Cre/wt}Rosa26^{DTA/wt}$) genotypes, if no lethality occurs. Litters were scored for phenotype (normal growth, growth retardation, resorbtions) from E9.5 to E13.5.

**Analysis of embryonic peripheral blood**. E11.5- E12.5 embryos were carefully removed from decidua and placenta, and washed three times in PBS. Upon placing intact embryos in individual wells containg 0.02% EDTA in PBS, umbilical vessels were cut and embryos were bled out. Cell suspensions were collected and spun for subsequent analyses (counting and FCM analysis).

**May Grunwald Giemsa staining of frozen sections and blood smears**. In total, 8-µm thin cryosections prepared from E12.5 FLs, and smears from PB were stained with May Grunwald Giemsa (MGG) stain (Diapath) according to the manu-facturer's intructions and imaged with a Leica DM6000 microscope equipped with a 40x objective. Images were processed in ImageJ.

**Transplantation of Tlr2-labeled BM cells into lethally irradiated mice**. $Rosa26^{EYFP}Tlr2^{CreERT2}$ embryos were pulsed with 4-OHT at E8.5. After 16 weeks, lineage-depleted BM (CD45.2) was transferred to lethally irradiated ($2 \times 7.5$ Gy delivered by orthovoltage X-ray instrument T-200; Wolf-Medizintechnik) reci-pients (CD45.1/2) along with support ($0.5 \times 10^6$ CD45.1) BM cells. Recipient mice were maintained on antibiotic water (gentamycin, 1 mg/ml) for 10 days. Sixteen weeks after primary cell transfer, BM was isolated and after lineage depletion, $Tlr2$EYFP+ cells were sorted and transferred to secondary lethally irradiated recipients (CD45.1/2) along with support ($0.5 \times 10^6$ CD45.1) BM cells. The per-centage of EYFP labeled cells and their contribution to CD45+ hematopoietic lineages in different organs was determined 16 weeks after induction or BM transfer, respectively.

**Whole-body imaging**. The total fluorescence of tdTomato in adult $Rosa26^{tdTomato}$ $Tlr2^{Cre}$ animals was acquired by Xtreme whole body imager (Bruker). Fluorescence spectra of excitation wavelengths from 420 nm to 540 nm with 580–629 nm emission filter were measured. Multispectral analyzer software (Bruker) was used to distinguish non-specific body autofluorescence from the specific tdTomato signal.

**Quantification and statistical analysis**. Data was statistically analyzed using Prism 5.03 software (GraphPad). The statistical tests used are indicated in the corresponding Figure legends.

**Reporting summary**. Further information on research design is available in the Nature Research Reporting Summary linked to this article.

## Data availability

The authors declare that all data supporting the findings of this study are available within the article and its supplementary information files or from the corresponding author upon reasonable request. The source data underlying Figs. 1, 3, 4, 5, 6 and 7, as well as Supplementary Figs. 1, 2, 3, 3e – heatmap, 4, 5 and 6 are provided as a Source Data file.

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

## Acknowledgements

We thank J. Manning for help with preparation of the paper, Z. Cimburek for FACS sorting, L. Klein for providing the ST-2 cells, J.C. Zuniga-Pflucker for OP-9 cells, Z. Kozmik for the *Cre* cassette, and Petr Bartunek for help with the colony assay. This work was supported by Grant 19–23154 S received by Grant Agency of the Czech Republic (GACR). I.S. was supported by Grant No. 200815 received from Grant Agency of Charles University (GAUK); M.A.J. by RVO 68378050; R.S. by grants LM2015040, and LQ1604 from the Ministry of Education, Youth and Sports of the Czech Republic (MEYS) and OP RDI CZ.1.05/1.1.00/02.0109 and CZ.1.05/2.1.00/19.0395 from the MEYS and European Regional Development Fund; K.F. by Grant Nr. 15-28525A from the Ministry of Health of the Czech Republic.

## Author contributions

D.F. and J.B. designed and supervised the experiments, wrote the paper. J.B., I.S., and M. D. performed experiments. M.K. and K.F. performed bioinformatic analyses. V.K. provided reporter mouse strains and *Cre*<sup>ERT2</sup> cassette, and M.A.J. supervised the BM transfer experiments. J.P. performed whole body imaging. J.B. and R.S. generated mouse *Tlr2*-transgenic models. I.G. designed and supervised colony assay and adviced on multiple experiments. A.J and H.S. performed two-photon microscopy imaging.

## Competing interests

The authors declare no competing interests.
