## [Peer Review File · Nature Communications]

Reviewers' Comments:

Reviewer #1:

Remarks to the Author:

In this paper, the authors use TLR2 expression to analyze the emergence of the hematopoietic system in staged mouse embryos. This work builds on the concept that 2 waves of hematopoietic progenitors emerge in the yolk sac prior to, but overlapping with, the emergence of pre-HSCs in the embryo. The overall conclusion of the paper, that TLR2 expression marks the emergence of EMP and pre-HSC/HSC appears sound overall. Extensive primary and supplementary data are provided to support these conclusions. A functional role for TLR2 is also shown, as TLR2 activation led to increased proliferation and myeloid differentiation of TLR2+Kit+ progenitors. An important strength of this paper is the generation and use of three TLR2 murine genetic models to lineage trace TLR2 expressing cells and to delete TLR2. Transplantation studies of labeled cells provide strong functional evidence for the expression of TLR2 in HSCs, with secondary transplantation nicely confirming this result.

However, several issues should be addressed.

A general concern exists regarding the lack of specific embryo staging. As the authors are well aware, developmental events in mouse embryos occur with surprising rapidity and reproducibility. While there are several instances where the somite counts of E8.5 and E9.5 embryos are listed (e.g. Fig. 2A), the specific staging of E7.5 embryos needs to be more carefully addressed. The attached diagrammatic figure implies that all of the E7.5 embryos were at the neural plate stage, while E7.5 embryos can vary from primitive streak to headfold stages within the same litter. These data can impact the interpretation of the relationships of primitive and definitive hematopoietic lineages.

While clonogenic assays were used to analyze the TLR2+Kit+ and TLR2-Kit+ cells (Figs. 1F/1G), the lack of mixed colonies in the TLR2-Kit+ cells was inferred to identify them as primitive erythroid/megakaryocyte progenitors. It is not clear that "primitive" hematopoietic progenitors express significant Kit on their cell surface. In addition, primitive erythroid colony formation peaks at day 5 of culture but would not be expected at day 12 of culture. Alternatively, the TLR2-Kit+ cells could represent EMP with restricted (less myeloid) lineage potential. The primitive vs definitive nature of these progenitors needs to be defined. One unequivocal approach would be to analyze globin gene expression of the erythroid colonies since the globin expression patterns are strikingly different between primitive and definitive erythroid lineages.

TLR2+/EYFP mice were used to track TLR2 expression during murine embryogenesis. It is surprising that at E8.5 the Kit-Ter+ population is labeled since these are likely primitive erythroid cells (sFig. 3A). If this is the case? Given the predilection of TLR to expression in myeloid lineage-biased cells, it is surprising that erythroid lineage cells are the predominant cell type labeled early.

The TLR2+/EYFP labeling of Kit-FcgR+ Mf at E8.5 (sFig. 3C), could be cells rapidly differentiating from EMP (as conjectured), however, they could be primitive macrophages independently labeled.

The deletion of TLR2-expressing cells using diphtheria toxin results in embryonic lethality at E13.5 (data not shown). What is the cause of the embryo lethality? Do TLR2-deleted embryos show any evidence of bleeding in the head or of severe anemia? While the absence of microglia is ascribed to the loss of EMP in these deletion experiments, an alternative explanation is that TLR2 is expressed not only in EMP, but also in primitive macrophages. I wonder whether the kit-FcgR+ (Mf) at E8.5 in Fig. 3A are primitive macrophages, partially labeled in the TLR2+/EYFP mice.

Pulse labeling of inducible TLR2 at E8.5 labels early fetal liver Kit+ cells, consistent with the labeling of EMP. Interestingly, the authors find that E8.5 tamoxifen pulsing at E8.5 also labels

adult hematopoietic cells, c/w the labeling of pre-HSC. Interpretation of the timing of labeling in these tracking data need to be tempered by the recent publication of the Medvinsky lab regarding the persistence of tamoxifen effects 72 hrs post exposure (Senserrich, 2018).

Minor issues:

1. A number of abbreviations need to be clarified in the text, including "MFs" (page 4), "PPP" (page 8), and "LMP" (page 13).

2. Page 4, top paragraph: It is not clear that primitive erythroid and megakaryocyte progenitors are not found earlier than monopotent macrophage progenitors in the murine embryo.

3. Page 4, top paragraph, last sentence is not precise enough. Yes, hematopoietic progenitors (EMP) seed the fetal liver as it forms, but it is likely that HSCs provide the progenitors that seed the BM, spleen, and thymus. References 11 and 12 are focused on HSC numbers in the murine embryo.

4. Page 4, bottom paragraph: primitive erythropoiesis is not completely independent of Runx1, since Runx1-null primitive erythroblasts are abnormal (Yokomizo, Blood, 2008).

5. Page 10 and Figure 3A, right panels: the authors identify Kit+FcgR+ cells as MfP and Kit+CD41+ cells as MKP. Colony assays are needed to verify the specific and restricted potential of these progenitors.

Reviewer #2:

Remarks to the Author:

Balounova et al. describe the expression of TLR2 during embryonic hematopoiesis. The authors use antibodies to endogenous TLR2 protein, and create a number of TLR2 mouse reagents (TLR2-Cre bred to Rosa reporters and DTA, and TLR2-CreERT) to analyze TLR2 expression, perform TLR2-Cre mediated lineage tracing, and to deplete TLR2 expressing cells. Their bottom line is that TLR2 is expressed on yolk sac-derived erythro-myeloid progenitors and TLR2-Cre marked cells give rise to adult HSCs.

The work is descriptive in nature. Furthermore, the manuscript contains so much data it is a bit tedious to read. Some of the data recapitulate already described features of embryonic hematopoiesis. For example, a long section on pulsed labeling of cells in the embryo using TLR2-CreERT and following their fate into the adult yields no new information about embryonic hematopoiesis. Furthermore, timed tamoxifen-activated CRE labeling is notoriously difficult to interpret, since the kinetics of CreERT activity (particularly how long it lasts) is hard to define. Nevertheless, there are a few interesting messages in the manuscript. One is the suggestion that TLR2, an inflammatory receptor, might be involved in EHT in two discrete anatomic sites, the yolk sac and caudal hematopoietic tissue. This would confirm and extend previous reports that inflammatory signals are required for hematopoietic stem and progenitor cell emergence (Espin-Palazon et al., Cell, 2014, He et al., Blood, 2015, Sawamiphak et al., Developmental Cell, 2014, and ref. 30). Another interesting message is that TLR2 expression may be able to discriminate between precursors of primitive erythrocytes/megakaryocytes and precursors of EMPs as early as E7.5. A more tightly focused manuscript demonstrating this point would probably generate more interest.

Specific comments:

1. I would suggest trimming down the figures by about 30%.

2. Page 8, results. One of the authors' conclusions is that TLR2-Kit+ progenitors are linked to primitive erythro/megakaryocytic progenitors. They should do a globin expression analysis to conclude that is the case. A similar analysis should be done for the TLR2+Kit+ cells to show that they produce definitive erythrocytes.

3. The authors did not show whether bona fide hemogenic endothelial express TLR2. There was some suggestion of this (i.e. Figure 1D and Figure 5), but it was not discussed or investigated at great length.

4. Page 16, line 4. Careful analysis of EMPs in functional assays by the Palis lab describe their appearance first at E8.25, as acknowledged by the authors. The E7.5 c-Kit+TLR2+ cells described by these authors to be EMPs could be precursors of EMPs, but according to published progenitor data there are no functional EMPs at that time. The authors need to be careful about describing their TLR2+Kit+ cells to be precursors of EMPs, and not EMPs.

5. Statistics needed for Fig. S1D.

6. The authors do not state in any section (main text, figure legend, materials and methods) when they administered DT. Knowing the timing is critical for interpretation of the results.

7. There is some contribution of TLR2-Cre marked embryonic cells to adult hematopoiesis. However, this contribution seems to be minimal. The best contribution of their labeling to adult hematopoiesis is less than 1% of the CD45+ fraction of peripheral blood, indicating that the bulk of adult hematopoiesis is coming from TLR2- cells. This detail is glossed over.

Minor points

8. Fig 1D and 1F should have consistent arrangement of the sorted populations since they are the same populations.

9. Fig 2 B/C: The authors' gating strategy as depicted does not demonstrate the existence of TLR2+CD11b+CD45+ cells. They need to show that the TLR2+CD11b+ population is CD45+ using hierarchical gating.

Reviewers' comments:

Reviewer #1 (Remarks to the Author):

Major concerns:

1. A general concern exists regarding the lack of specific embryo staging. As the authors are well aware, developmental events in mouse embryos occur with surprising rapidity and reproducibility. While there are several instances where the somite counts of E8.5 and E9.5 embryos are listed (e.g. Fig. 2A), the specific staging of E7.5 embryos needs to be more carefully addressed. The attached diagrammatic figure implies that all of the E7.5 embryos were at the neural plate stage, while E7.5 embryos can vary from primitive streak to headfold stages within the same litter. These data can impact the interpretation of the relationships of primitive and definitive hematopoietic lineages.

We agree that this is indeed a crucial point to our claim that precursors of EMP appear in embryos at ~E7.5. We staged embryos according to the guidelines set forth by K.M. Downs and T. Davies in 1993 which specifically deal with staging of gastrulating mouse embryos. In all relevant experiments, we strictly used embryos from E7.5 time pregnant females that were at the neural plate stage (NPS) and younger, with no visible signs of allantoic bud or structure and with the apparent presence of the cranial limiting furrow and pointy node. All embryos, which were older than those that presented as „no allantois bud stage“ of NPS, were excluded from CFU assays experiments. We are confident that the age range of the vast majority of embryos were between E7,25-7,5 (on a rare occasion may include up to 7,75). For illustration, we have attached a picture which documents such selection process of embryos for colony assay experiments. As seen in Fig.1-2, all embryos from two litters which were examined as well as those which failed to fulfill the above criteria, were excluded (marked with a red cross). Usually, the images of embryos which were selected for each experiment were recorded. Fig.2-3 show examples of these final images for two individual independent experiments.

For simplification of the figure agenda, we used a diagrammatic cartoon that represents a typical E7.5 embryo. To clarify how E7.5 embryos were staged and selected, we also added this information to the Method section of the Ms, the subsection „Animals“.

2. While clonogenic assays were used to analyze the TLR2⁺Kit⁺ and TLR2-Kit⁺ cells (Figs. 1F/1G), the lack of mixed colonies in the TLR2-Kit⁺ cells was inferred to identify them as primitive erythroid/megakaryocyte progenitors. It is not clear that “primitive” hematopoietic progenitors express significant Kit on their cell surface. In addition, primitive erythroid colony formation peaks at day 5 of culture but would not be expected at day 12 of culture. Alternatively, the TLR2-Kit⁺ cells could represent EMP with restricted (less myeloid) lineage potential. The primitive vs definitive nature of these progenitors needs to be defined. One unequivocal approach would be to analyze globin gene expression of the erythroid colonies since the globin expression patterns are strikingly different between primitive and definitive erythroid lineages.

We would like to thank the reviewer for pointing out this testable prediction. Indeed, in the colony forming assay, c-kit⁺TLR2⁻ progenitors generated erythroid colonies with a significantly higher ratio of embryonic (Hbb-bh1) to adult (Hbb-b1) globin genes than their TLR2⁺c-kit⁺ counterparts. This new data is presented in the revised **Suppl. Fig.2b**. Moreover, two additional sets of data equally point to the distinct nature of c-kit⁺TLR2⁻ and TLR2⁺c-kit⁺ progenitors. First, *Tlr2*-driven EYFP labeling of primitive erythrocytes (EryP) was inefficient when compared to EMPs (**Suppl. Fig.4e**). Second, and consistent with their inefficient EYFP labelling, EryPs with their typical morphology and phenotype were not significantly affected upon *Tlr2*-mediated cell depletion in E12.5 *Tlr2^{Cre}Rosa26^{DTA}* embryos (**Suppl Fig.6d and 6e**). On the other hand, EMPs in the peripheral blood and liver of the very same tg embryos had nearly vanished. This data argues against the scenario considered by the reviewer that c-kit⁺TLR2⁻ and c-kit⁺TLR2⁺ cells represent progenitors of primitive versus definitive transient (EMPs) YS hematopoiesis, respectively.

3. TLR2⁺/EYFP mice were used to track TLR2 expression during murine embryogenesis. It is surprising that at E8.5 the Kit-Ter⁺ population is labeled since these are likely primitive erythroid cells (sFig. 3A). Is this the case? Given the predilection of TLR to expression in myeloid lineage-biased cells, it is surprising that erythroid lineage cells are the predominant cell type labeled early.

While EryPs were labelled already in E8.5 and E9.5 *Tlr2^{Cre}Rosa2^{EYFP}* embryos and represented approx. 40% of all *Tlr2-Cre* labelled cells (**Suppl. Fig.4a, upper row, scatter plot**), these labelled EryPs represented an insignificant fraction, notably 0.73%±0.307 and 3.1%±0.97, respectively, of all EryPs (**Fig.3a, two upper rows, scatter plots, EryP**). In fact, such low labeling efficiency of EryP was well below the labeling threshold (average labeling efficiency of all live cells) (**Fig.3a, scatter plots**). On the other hand, EMPs, at E8.5 and E9.5 labelled on average 61,8%±7,3 and 84,09%±2,16, respectively, of all *Tlr2-Cre* labelled cells (**Fig.3a, scatter plots**). This data suggests that while the *Tlr2* locus could be stochastically activated in all hematopoietic cells, including EryPs, its deliberate activation is predominantly targeted to the precursors of EMPs at developmental time window E7.5-8.5 when these precursors start to appear and can be thus readily detected as TLR2⁺c-kit⁺ cells.

4. The TLR2⁺/EYFP labeling of Kit-FcγR⁺ Mf at E8.5 (sFig. 3C), could be cells rapidly differentiating from EMP (as conjectured), however, they could be primitive macrophages independently labeled.

We thank the reviewer for pointing to this ambiguous statement in our Ms. In our original Ms, we considered E8.5 *Tlr2*-labeled c-kit⁻ FcRγ⁺ cells as differentiated embryonic MFs, because they were of the c-kit-FcRγ⁺ phenotype, i.e. similar to that of c-kit-FcRγ⁺ maternal MFs (mMFs) which were present

in E8.5 embryo (**Suppl. Fig.4c**, originally Suppl. Fig.3c). However, as our current analysis showed, these FcR γ + embryonic cells are not only Ter19 $^-$ -c-kit $^-$ CD41 $^-$ but also negative for CD11b and CD45, and thus they can't be formally classified even as hematopoietic cells. In addition, they represent only a miniscule fraction (on average only 0.33%) of all Tlr2 Cre tdTomato $^+$ recombined cells in E8.5 embryos. So, based on this analysis, it is impossible to make any statement about their origin and affiliation to a certain embryonic hematopoietic wave and/or lineage. For this reason, we have changed the label of this subset from MFs to embryonic FcR γ + cells (eFcR γ) and added a new explanatory statement to the text of the Ms.

5. The deletion of TLR2-expressing cells using diphtheria toxin results in embryonic lethality at E13.5 (data not shown). What is the cause of the embryo lethality? Do TLR2-deleted embryos show any evidence of bleeding in the head or of severe anemia?

The deletion of cells with an active *Tlr2* locus in the *Tlr2 Cre Rosa26 DTA* model leads to embryonic lethality. We were unable to find any viable pups at P0 with a *Tlr2 Cre Rosa26 DTA* genotype (n=7, data not shown). On rare occasions, *Tlr2 Cre Rosa26 DTA* embryos were found in few E13.5 litters. Detailed analysis of E9.5-E13.5 litters has revealed that the first resorptions of *Tlr2 Cre Rosa26 DTA* embryos had already occurred at E10.5 and peaked at E13.5. From E10.5 to E11.5, we found significant numbers of developmentally retarded (usually smaller and pale) *Tlr2 Cre Rosa26 DTA* embryos (**see the revised Suppl. Fig.6b**). Flow cytometry (FCM) analysis revealed in these embryos many dead or dying cells detected by increased autofluorescence. Therefore, only those *Tlr2 Cre Rosa26 DTA* embryos were considered for subsequent comparative and FCM analyses which were of similar size and morphology to those of the *Tlr2 wt Rosa26 DTA* genotype. It is important to emphasize that, especially in E12.5 litters, *Tlr2 Cre Rosa26 DTA* embryos with similar size and morphology to *Tlr2 wt Rosa26 DTA* embryos represented only about one half of *Tlr2 Cre Rosa26 DTA* embryos, the rest being already resorbed. These embryos survived till E12.5 probably due to a weaker or delayed *Tlr2*-driven recombination of the DTA cassette. Since no viable *Tlr2 Cre Rosa26 DTA* pups at P0 (n=7) or E17.5 embryos (n=5) were found, we assumed that all *Tlr2 Cre Rosa26 DTA* embryos which were still viable at E12.5 were heading toward death. Nevertheless, when E12.5 *Tlr2 Cre Rosa26 DTA* embryos were analyzed they showed a dramatic aberration of EMP driven hematopoiesis with no impact on primitive erythropoiesis (**see the revised Suppl. Fig.6d and e**). Notably, surviving E12.5 *Tlr2 Cre Rosa26 DTA* embryos showed no evidence of bleeding and/or paleness, with normal counts of nucleated EryPs in the blood stream, suggesting normal development of EryPs (**Suppl. Fig.6d**). However, their FLs were pale and, importantly, FL EMPs with their progeny (FL MF, Early erythroid progenitors, Ery) were substantially decreased or depleted from E12.5 FLs and blood stream. In contrast, while the cellularity of FLs was also decreased, non-hematopoietic cell counts (Ter119- c-kit- CD45-) did not differ among *Tlr2 wt Rosa26 DTA* and *Tlr2 Cre Rosa26 DTA* FLs (**Suppl. Figure 6d**). Since at E12.5 FL erythropoiesis takes over primitive erythropoiesis (reviewed in (Fraser, 2013)) and the primitive erythropoiesis was not affected in *Tlr2 Cre Rosa26 DTA* embryos, this data argues in favor of the scenario whereby the depletion of definitive hematopoiesis strongly contributed to the observed lethality at E13.5 in *Tlr2 Cre Rosa26 DTA* embryos.

This new set of data was added to the text of the Ms and to revised **Suppl. Fig.6d and e**.

While the absence of microglia is ascribed to the loss of EMP in these deletion experiments, an alternative explanation is that TLR2 is expressed not only in EMP, but also in primitive macrophages. I wonder whether the kit-FcgR+ (Mf) at E8.5 in Fig. 3A are primitive macrophages, partially labeled in the TLR2+/EYFP mice.

This concern is related to question #4. Since c-kit-FcR γ ⁺ cells can't be considered to be hematopoietic/myeloid cells, (lack of CD45, as well as the myeloid marker CD11b), the question related to the relationship between primitive macrophages and microglia can't be directly addressed in this experimental setting. At this stage of development, until E9.5, there are virtually no CD45⁺ embryonic cells (Fig. 1e) which could be considered as differentiated mature primitive MFs.

Pulse labeling of inducible TLR2 at E8.5 labels early fetal liver Kit⁺ cells, consistent with the labeling of EMP. Interestingly, the authors find that E8.5 tamoxifen pulsing at E8.5 also labels adult hematopoietic cells, c/w the labeling of pre-HSC. Interpretation of the timing of labeling in these tracking data need to be tempered by the recent publication of the Medvinsky lab regarding the persistence of tamoxifen effects 72 hrs post exposure (Senserrich, 2018).

This is also a very important issue to which we paid an extra attention regarding the interpretation of our data. The work of Senserrich and colleagues (Senserrich et al., 2018) provided evidence that a significant Cre-inducing tamoxifen activity persists in mouse blood for at least 72 hours after injection. However, such prolonged persistence of tamoxifen activity in our system is not consistent with our results. The most compelling piece of evidence strongly arguing against such scenario is the data shown in **Fig.4b**. Specifically, while the injection of 4-hydroxytamoxifen (4-OHT) into *Tlr2^{CreERT2}Rosa26^{EYFP}* at E7.75 and E 8.25 lead to cell labelling in approx. 50% and 90% of embryos, respectively, i.e. when EMPs emerge, it caused no labelling whatsoever when administered 6 hours prior, at E7.5. If the 4-OHT and/or its metabolites persists in concentrations that are required for a recombination a few hours more, we would see the labelling of these emerging EMPs also in E7.5. Similar logic can be used for the results highlighted in **Fig.4c**, **Suppl. Fig.5f** as well as for **Fig.7a**, the latter being directly related to the claim of specific labeling of emerging pre-HSCs at E8.5. Here again, had the tamoxifen activity persisted for ≥ 1 day, then efficient labeling of *bona-fide* LT-HSC pre-HSCs would also occur by the administration of 4-OHT between E7.5 and E8.25, which is not the case in our system. Moreover, given the extreme scarcity of HSCs and their precursors in \leq E10.5 embryos, it is highly unlikely that their recombination would be activated by the significantly diminished activity of 4-OHT after 20 hours. There are several reasons for such a discrepancy with the above indicated report by Senserrich et al., 2018. First, we used a lower dose of 4-OHT (1.5mg versus 2.0mg) for the induction of recombination. Second, and in our opinion very important, the experimental conditions for testing 4-OHT activity, as described by Senserrich and colleagues, are quite distinct from those used in our model. In that study, the authors injected 2mg of 4-OHT intraperitoneally into wild-type C57BL/6 females and were able to detect tamoxifen activity *in vitro* in the serum taken from these mice up to 72 hours upon administration. However, as the negative control in this experiment is omitted (Fig.3 in Senserrich et al., 2018), largely unchanged levels of tamoxifen activity between 20 and 72 hrs post administration (reaching approx. 20-50% of its maximal activity at 5 hr) do not allow distinction whether this is background or indeed the persistence of its activity). The highest tamoxifen activity, however, was observed 5 hours upon its administration. In contrast, we injected 4-OHT in peritoneum of wt *Rosa26^{EYFP}* females carrying *Tlr2^{CreERT2}Rosa26^{EYFP}* embryos, where 4-OHT had the opportunity to bind instantly to readily available Cre-protein-fused-estrogen receptors, thus being rapidly eliminated from the system. Third, and consistent with previously published data using the *Tie2* pulse-labeling model (Gomez Perdiguero et al., 2015), a single dose of 4-OHT used from E7.5 to E10.5 in *Tlr2^{CreERT2}Rosa26^{EYFP}* animals resulted in a relatively low labeling efficiency of target cells (0.1-2% of total cells).

For the above reasons, as well as for the fact that we were unable to find conclusive evidence for prolonged tamoxifen activity in our system, we are confident, that our inducible cell-labeling data is not a consequence long-lasting activity of 4-OHT, but rather reflect short term activity of this inducer of recombination/cell labelling in our system.

Minor issues:

1. A number of abbreviations need to be clarified in the text, including "MFs" (page 4), "PPP" (page 8), and "LMP" (page 13).

All these abbreviations were explained in text of the original Ms. Notably, MFs (macrophages), EMPs (erythro-myeloid progenitors) and LMPs (lympho-myeloid progenitors) are introduced in the first paragraph of the Intro section, page 4. We assume that the acronym "PPP" which was noted actually refers to PPS (posterior primitive streak), which was introduced in the first paragraph of the result section.

2. Page 4, top paragraph: It is not clear that primitive erythroid and megakaryocyte progenitors are not found earlier than monopotent macrophage progenitors in the murine embryo.

In our Introduction, we provide a literature overview highlighting when the progenitors of cells of the first wave emerge, whereby all statements are accompanied by the relevant references, quote: "The first, referred to as the primitive wave, arises in the yolk sac (YS) at 7.25 and consists of progenitors of primitive nucleated erythrocytes (Ery-P) and megakaryocytes (Mk)(Ferkowicz et al., 2003; Tober et al., 2007; Xu et al., 2001). Monopotent progenitors of macrophages (MFs), which appear shortly thereafter, are also considered to be part of this wave (Bertrand et al., 2005; Palis et al., 1999)". With all respect, we are confident, that these statements reflect the overall consensus in research community in this field.

3. Page 4, top paragraph, last sentence is not precise enough. Yes, hematopoietic progenitors (EMP) seed the fetal liver as it forms, but it is likely that HSCs provide the progenitors that seed the BM, spleen, and thymus. References 11 and 12 are focused on HSC numbers in the murine embryo.

We agree with the reviewer that this statement could be more accurate. We have rephrased the sentence to state: "Commencing from E9.5, EMPs from YS and other hemogenic tissues, and at E10.5 HSCs from AGM, start to seed and expand in the forming fetal liver (FL). Subsequently, these progenitors migrate and differentiate to tissue-resident MFs, with the latter populating the bone marrow (BM), spleen, or thymus."

4. Page 4, bottom paragraph: primitive erythropoiesis is not completely independent of Runx1, since Runx1-null primitive erythroblasts are abnormal (Yokomizo, Blood, 2008).

In their study, Yokomizo et al (Blood, 2008), showed that about 27% of EryP in *Runx1* deficient embryos display an abnormal shape, however, they are still fully functional, since they exhibit normal hemoglobinization, don't cause lethality *per se*, and the number of primitive erythroid colonies derived from these embryos were similar to those derived from WT embryos. Thus, indeed, the primitive erythropoiesis is functionally independent of *Runx1*. Nevertheless, we rephrased the sentence referring to the role of *Runx1* in primitive hematopoiesis: "This is in agreement with the observation, that in *Runx1* deficient embryos primitive erythropoiesis occurs, but EMPs, HSCs, and also MFs are absent..."

5. Page 10 and Figure 3A, right panels: the authors identify Kit+FcgR+ cells are MfP and Kit+CD41+ cells as MKP. Colony assays are needed to verify the specific and restricted potential of these progenitors.

We appreciate this suggestion. However, our intention is to focus the manuscript on definitive hematopoiesis, EMPs and HSCs in respect to TLR2, utilized here as the marker of their progenitors. Experiments which are beyond the scope of our Ms have been so far omitted and likely will be the subject of further studies.

Reviewer #2 (Remarks to the Author):

Specific comments:

1. I would suggest trimming down the figures by about 30%.

Thanks for this suggestion. Based on this recommendation and to make our Ms more “breatheable”, we trimmed the main figures. Specifically, the original figures No. 2, 4, 5, 6 and 7 were trimmed and/or simplified. We believe that the current version of figures makes the Ms easier to follow.

2. Page 8, results. One of the authors' conclusions is that TLR2-Kit+ progenitors are linked to primitive erythro/megakaryocytic progenitors. They should do a globin expression analysis to conclude that is the case. A similar analysis should be done for the TLR2+Kit+ cells to show that they produce definitive erythrocytes.

Thank you for this suggestion. As described in the answer to the question #2, reviewer #1, we have performed this analysis and confirmed that the hemoglobin expression profile is consistent with the notion that TLR2-Kit+ and TLR2+Kit+ cells represent progenitors of primitive versus definitive transient hematopoiesis, respectively.

3. The authors did not show whether bona fide hemogenic endothelial express TLR2. There was some suggestion of this (i.e. Figure 1D and Figure 5), but it was not discussed or investigated at great length.

We appreciate this question. However, while the *Tlr2* locus is clearly activated in endothelial cells and IAHCs of the dorsal aorta (Fig.6), the level of TLR2 surface expression can be too low and hence below the detection by anti-TLR2 antibody. Specifically, the surface expression of TLRs is notoriously known to be low and accompanied by difficulties by direct staining with TLR-specific antibodies, especially on fixed histological slices. Our attempt to performed this analysis was inconclusive, likely due to relatively low levels of surface TLR2 on endothelial cells.

4. Page 16, line 4. Careful analysis of EMPs in functional assays by the Palis lab describe their appearance first at E8.25, as acknowledged by the authors. The E7.5 c-Kit+TLR2+ cells described by these authors to be EMPs could be precursors of EMPs, but according to published progenitor data there are no functional EMPs at that time. The authors need to be careful about describing their TLR2+Kit+ cells to be precursors of EMPs, and not EMPs.

We are very thankful for this suggestion and concur with the reviewer that E7.5 c-Kit+TLR2+ cells represent the progenitors of EMPs. Notably, because the cells isolated at E7.5 do not exhibit the typical phenotype of EMPs (c-kit+CD16/32+CD41+) which can be found in the embryo starting at E8.5, but at

E7.5 are c-kit+TLR2+CD16/32-CD41-, they represent the precursors of EMPs which start to retain CD41 and FcR γ 24 hrs later. We corrected this inaccuracy in the manuscript.

5. Statistics needed for Fig. S1D.

Thank you for the notification. We added the statistics for the **Fig.S1D**.

6. The authors do not state in any section (main text, figure legend, materials and methods) when they administered DT. Knowing the timing is critical for interpretation of the results.

In the experiments, where the cells with active *Tlr2* locus were deleted, the mouse strain with constitutive expression of Cre from *Tlr2* regulatory elements (*Tlr2*^{Cre}) was used (**Fig.5** in the revised Ms). When *Tlr2*^{Cre} mice is crossed to *Rosa26*^{DTA} reporter strain, Cre recombinase activates the DTA module, the toxin is produced and directly induces cell death.

7. There is some contribution of TLR2-Cre marked embryonic cells to adult hematopoiesis. However, this contribution seems to be minimal. The best contribution of their labeling to adult hematopoiesis is less than 1% of the CD45+ fraction of peripheral blood, indicating that the bulk of adult hematopoiesis is coming from TLR2- cells. This detail is glossed over.

Similaral to other genetic pulse-labeling models, when a single dose of 4-OHT is used, the marking efficiency is relatively low (0.1-2% of total cells) (Gomez Perdiguero et al., 2015; Busch et al., 2015). This is consistent with the fact, that tamoxifen activity is time limited as is the number of target cell at the time of tamoxifen administration. However, when one looks at the mouse strain with constitutive expression and labeling of progenitors (*Tlr2*^{Cre}*Rosa26*^{EYFP}, **Suppl.Fig. 3**), the labeling efficiency of CD45⁺ hematopoietic cells in peripheral organs such as spleen, BM, LN and blood is nearly 100% (**Suppl. Fig.3c**).

Minor points

8. Fig 1D and 1F should have consistent arrangement of the sorted populations since they are the same populations.

As suggested, we have changed the arrangement of these figures.

9. Fig 2 B/C: The authors' gating strategy as depicted does not demonstrate the existence of TLR2+CD11b+CD45+ cells. They need to show that the TLR2+CD11b+ population is CD45+ using hierarchical gating.

We appreciate this concern. While there would be no problem to show hierarchical gating, due to trimming down the figures, **Fig.2b and c** were removed, as the **Fig.3** provides all necessary information related to the emergence of TLR2-labelled cells and their affiliation to hematopoietic lineages.

Reviewers' Comments:

Reviewer #1:

Remarks to the Author:

The authors have responded adequately to all of the concerns raised during the initial review, with significant alterations to the text, some streamlining of the figures and the addition of supplementary data. The overall conclusion that TLR2 is expressed by the emerging EMP and HSC lineages of hematopoiesis is further strengthened.

The only remaining, and extremely minor, concern is the statement in the Introduction that primitive erythroid and megakaryocyte progenitors are found earlier than monopotent macrophage progenitors. As referenced by the authors, Figure 1 in Palis, et al. 1999 indicates that both primitive erythroid and macrophage progenitors are present in mid-streak mouse embryos. Likewise, in Fig. 2C in Tober, et al. 2007, primitive erythroid/megakaryocyte bipotential progenitors are also first identified at mid-late primitive streak stages. While it is clear that maturing macrophage cells are detected later than primitive erythroblasts, I am not aware of any published data indicating that primitive erythroid progenitors emerge before unipotent macrophage progenitors.

Reviewer #2:

Remarks to the Author:

The authors have adequately addressed the reviewers' comments.

Reviewers' comments:

Reviewer #1 (Remarks to the Author):

The only remaining, and extremely minor, concern is the statement in the Introduction that primitive erythroid and megakaryocyte progenitors are found earlier than monopotent macrophage progenitors. As referenced by the authors, Figure 1 in Palis, et al. 1999 indicates that both primitive erythroid and macrophage progenitors are present in mid-streak mouse embryos. Likewise, in Fig. 2C in Tober, et al. 2007, primitive erythroid/megakaryocyte bipotential progenitors are also first identified at mid-late primitive streak stages. While it is clear that maturing macrophage cells are detected later than primitive erythroblasts, I am not aware of any published data indicating that primitive erythroid progenitors emerge before unipotent macrophage progenitors.

We thank the reviewer for this remark. We have changed the sentence accordingly: “The first, referred to as the primitive wave, arises in the yolk sac (YS) at 7.25 and consists of progenitors of primitive nucleated erythrocytes (EryP) and megakaryocytes (Mk)^{1,2,3}. Monopotent progenitors of macrophages (MFs) are also considered to be part of this wave^{4,5}.”